# Controlled Double-Direction Cyclic Quantum Communication of Arbitrary Two-Particle States

**DOI:** 10.3390/e27030292

**Published:** 2025-03-11

**Authors:** Nueraminaimu Maihemuti, Zhanheng Chen, Jiayin Peng, Yimamujiang Aisan, Jiangang Tang

**Affiliations:** 1School of Mathematics and Statistics, Kashi University, Kashi 844000, China; nuer0413@163.com (N.M.); greatxyzme@163.com (Y.A.); jg-tang@163.com (J.T.); 2The School of Mathematics and Statistics, Yili Normal University, Yili 835000, China; czh918czh@163.com

**Keywords:** CDDC, controlled cyclic QT, controlled cyclic RSP, 25-particle entangled state

## Abstract

With the rapid development of quantum communication technologies, controlled double-direction cyclic (CDDC) quantum communication has become an important research direction. However, how to choose an appropriate quantum state as a channel to achieve double-direction cyclic (DDC) quantum communication for multi-particle entangled states remains an unresolved challenge. This study aims to address this issue by constructing a suitable quantum channel and investigating the DDC quantum communication of two-particle states. Initially, we create a 25-particle entangled state using Hadamard and controlled-NOT (CNOT) gates, and provide its corresponding quantum circuit implementation. Based on this entangled state as a quantum channel, we propose two new four-party CDDC schemes, applied to quantum teleportation (QT) and remote state preparation (RSP), respectively. In both schemes, each communicating party can synchronously transmit two different arbitrary two-particle states to the other parties under supervisory control, achieving controlled quantum cyclic communication in both clockwise and counterclockwise directions. Additionally, the presented two schemes of four-party CDDC quantum communication are extended to situations where n>3 communicating parties. In each proposed scheme, we provide universal analytical formulas for the local operations of the sender, supervisor, and receiver, demonstrating that the success probability of each scheme can reach 100%. These schemes only require specific two-particle projective measurements, single-particle von Neumann measurements, and Pauli gate operations, all of which can be implemented with current technologies. We have also evaluated the inherent efficiency, security, and control capabilities of the proposed schemes. In comparison to earlier methods, the results demonstrate that our schemes perform exceptionally well. This study provides a theoretical foundation for bidirectional controlled quantum communication of multi-particle states, aiming to enhance security and capacity while meeting the diverse needs of future network scenarios.

## 1. Introduction

Quantum entanglement is essential for quantum communication, as it reveals non-local correlations between distant subsystems. QT and RSP are prime examples of quantum entanglement applications, both involving the transfer of quantum states between locations without the actual physical travel of the particles involved. QT was introduced by Bennett et al. [1] in 1993. It allows for the secure teleportation of any arbitrary unknown single-particle state from one location to another using pre-shared entanglement between two parties as the quantum channel, with the support of classical communication. Since then, various QT protocols have been proposed, such as controlled QT [2,3,4], bidirectional QT [5,6], controlled bidirectional QT [7,8], cyclic QT [9,10], and controlled cyclic QT [11,12,13], each enhancing aspects like security or communication capacity. Additionally, in 2000, Lo [14] introduced RSP, which can be viewed as a form of QT. Unlike QT, the prepared single- or multi-particle state does not need to be owned or measured by the transmitter. Instead, its coefficients are completely known to the transmitter, but not to the recipient at all. Therefore, RSP consumes less classical resources at the cost of the target state being completely transparent to the sender. Since its introduction, RSP has garnered significant attention, leading to the development of various enhanced RSP protocols, such as multicast-based multiparty RSP [15], controlled RSP [16,17], joint RSP [18,19,20,21,22], controlled joint RSP [23], and bidirectional controlled RSP [24,25,26,27], among others. However, these protocols are limited to one-way or two-way communication only.

In recent years, the development of quantum networks has brought new challenges and opportunities to the study of quantum communication. Quantum networks are regarded as a core infrastructure for achieving distributed quantum computing, ultra-secure communication, and high-precision quantum sensing. They are also an essential component of the future quantum internet [28,29,30]. Kimble first proposed the concept of the quantum internet in 2008 [28], emphasizing that quantum networks can enable global quantum information processing through entanglement distribution. With the continuous progress of technology, significant advancements have been made in the capacity of quantum networks [29], the computational efficiency of nonlinear Bell inequalities [30], and the feasibility of quantum repeaters [31]. These studies not only provide important theoretical and technical support for the design of quantum communication protocols but also highlight the pivotal role of quantum networks in the future of quantum information technology [32,33].

To address the demand for multiparty quantum communication in quantum networks, Chen et al. [34] proposed a novel cyclic QT scheme in 2017. This scheme utilizes a six-particle entangled state as the quantum channel to facilitate single-particle state transmission among multiple participants. Specifically, in this protocol, Alice teleports a single-particle state to Bob, Bob transmits a single-particle state to Charlie, and Charlie simultaneously conveys a single-particle state back to Alice, forming a closed cyclic communication loop. This scheme was further extended to scenarios involving *n* (n>3) participants, offering a flexible solution for multiparty quantum communication in quantum networks. Peng et al. [13] expanded on Chen’s work, introducing a framework for cyclic controlled QT where each participant, with the supervisor’s permission, teleports a state to neighbors. This led to the development of cyclic quantum communication protocols like controlled cyclic QT [35,36] and cyclic (controlled) RSP [37,38,39,40]. These protocols, while considering cyclic transmission in both directions, have a limitation: adjacent participants cannot exchange their quantum states, which may not fit real-world applications. More recently, Jiang et al. [41] proposed a hybrid dual-channel protocol, allowing communicators to transmit both known and unknown single-particle states to two others using RSP and QT, respectively. This protocol supports both cyclic RSP and QT communication in both directions, enhancing flexibility. Additionally, from a different viewpoint, adjacent communicators are able to swap their quantum states, which could significantly enhance the functionality of future quantum communication networks. Building on this, Sun and Zhang [42] introduced a DDC controlled RSP protocol for single-particle states, inspired by the single-particle quantum multicast concept, using a thirteen-particle entangled state as the quantum channel. The following year, they extended this idea by proposing a four-party scheme for implementing DDC controlled RSP for arbitrary two-particle states [43]. In this scheme, with the supervisor’s approval, each communicator can assist the other two communicators in preparing their respective two-particle target states simultaneously. In 2023, Peng et al. [44] further advanced the field by proposing a DDC controlled quantum communication scheme for single-particle states.

Although the existing studies have laid a solid foundation for multiparty quantum communication, further exploration of more efficient and flexible quantum resources and protocols is still needed to meet the increasingly complex and diverse demands of future quantum networks. To address this challenge, this paper proposes a scheme for constructing a 25-particle entangled state using Hadamard and CNOT gates as the quantum channel. Based on this entangled state, we designed DDC controlled QT and RSP protocols for arbitrary two-particle states. In the four-party scenario, the proposed protocol allows each communicator to simultaneously transmit two different two-particle states to the other two communicators, achieving cyclic communication in both clockwise and counterclockwise directions, significantly enhancing the flexibility and applicability of the quantum communication protocol. Moreover, the proposed protocol achieves a 100% success rate, with the required operations consisting only of two-particle projective measurements, single-particle von Neumann measurements, and Pauli gates, all of which can be implemented with current quantum technologies, ensuring experimental feasibility. We further extend the four-party protocol to an (n+1)-party scenario (n>3) and propose a new method for constructing an (8n+1)-particle entangled state to meet the demands of multiparty communication in large-scale quantum networks. While Peng et al. have already proposed a quantum channel and circuit design for single-particle states, the complexity and technical requirements increase significantly when extending the protocol to two-particle states. Compared to previous research, our work breaks through the limitations of single-particle states and is the first to apply the DDC quantum communication protocol to two-particle states. By constructing a new quantum channel, we successfully overcome the technical bottlenecks in two-particle state cyclic communication and propose a more efficient and scalable quantum communication model. This innovation not only expands the types of particles and transmission modes in quantum communication but also provides a solid theoretical foundation for the implementation of multi-particle quantum communication. Additionally, we provide more precise and practical mathematical formulas, offering important theoretical support for future research in multi-particle quantum communication. We also analyze the efficiency, security, and control capabilities of the scheme, and enhance the stability and flexibility of communication by optimizing multiparty control mechanisms. This offers solid support for the practical implementation and use of quantum communication networks. The results of this research will promote the widespread application of quantum communication technologies in large-scale quantum networks, further advancing the field of quantum information science.

The rest of this article is structured as follows. In Section 2, we suggest a method for constructing a multi-particle quantum channel using a 25-particle entangled state as an example, provide the detailed steps for implementing a four-party DDC controlled QT scheme of arbitrary unknown two-particle states, and extend this transmission scheme to (n+1)-party (n>3) via an (8n+1)-particle entangled state as the quantum channel. Section 3 describes the four-party DDC controlled RSP scheme by introducing auxiliary 12 particles, and promotes it to the scenario with *n* (n>3) communicators. Finally, conclusions are discussed and drawn in Section 4.

## 2. Double-Direction Cyclic Controlled Quantum Teleportation of Arbitrary Two-Particle States

We introduce a four-party DDC-controlled QT scheme with three communicators—Alice, Bob, and Charlie—and a supervisor, David.With the supervisor’s approval, each communicator can simultaneously teleport two distinct unknown two-particle states to the other communicators, while also receiving two unknown two-particle states from them. This setup allows for cyclic controlled QT in both clockwise and counterclockwise directions. Figure 1 illustrates the relationships among the three communicators and the controller, emphasizing the quantum states being teleported and the transmission of control information.

The detailed description of this scheme is provided in Section 2.2, and we extend the scheme to a general case involving multiple communicators, with the specific protocol and corresponding quantum circuit diagram presented in Section 2.3. To implement the proposed scheme, a 25-particle entangled state is required as the quantum channel, and the specific steps for constructing this quantum channel are detailed in Section 2.1.

### 2.1. Construction of the Quantum Channel

To implement the four-party DDC controlled QT scheme, we begin by constructing a 25-particle maximally entangled state to serve as the quantum channel. In Yu’s protocol [15], an eight-particle maximally entangled state, denoted as $|\varphi_{1}^{8}\rangle$, is used for the quantum multicast of two-particle states. Building on this eight-particle maximally entangled state, we extend it to a 25-particle maximally entangled state to fulfil the requirements for four-party controlled QT. Specifically, the quantum channel state is expressed as(1)$$\begin{array}{ccc} {|G\rangle }_{1,2,\cdots ,25} & =\frac{1}{\sqrt{2}}{(|\varphi_{1}^{8}\rangle }_{12\cdots 8}|\varphi_{1}^{8}\rangle_{9,10,\cdots ,16}|\varphi_{1}^{8}\rangle_{17,18,\cdots ,24}{|0\rangle }_{25} & \\ \; & \;\;\;+|\varphi_{2}^{8}\rangle_{12\cdots 8}|\varphi_{2}^{8}\rangle_{9,10,\cdots ,16}|\varphi_{2}^{8}\rangle_{17,18,\cdots ,24}{|1\rangle }_{25}), & \end{array}$$
where $|\varphi_{1}^{8}\rangle$ and $|\varphi_{2}^{8}\rangle$ are the eight-particle maximally entangled states, the explicit forms of which are provided in Appendix A as Equations (A1) and (A2).

Figure 2 shows the quantum circuit used to create the 25-particle maximally entangled state.

The Hadamard gate *H* and the CNOT gate are represented by the following matrices:(2)$$H=\frac{1}{\sqrt{2}}\left(\begin{array}{cc} 1 & 1\\ 1 & -1 \end{array}\right),\;\;\;CNOT=\left(\begin{array}{cccc} 1 & 0 & 0 & 0\\ 0 & 1 & 0 & 0\\ 0 & 0 & 0 & 1\\ 0 & 0 & 1 & 0 \end{array}\right).$$

We begin by preparing the 25-particle maximally entangled state from the initial state, where all qubits are in the |0〉 state. The first step is to apply a Hadamard gate *H* to the 25th qubit, which changes its state from |0〉 to a superposition state of $\frac{1}{\sqrt{2}}(|0\rangle +\left|1\rangle \right)$. The quantum state at this stage is(3)$$\begin{array}{c} |G_{1}\rangle_{1,2,\cdots ,25}={|00\cdots 0\rangle }_{1,2,\cdots ,24}⊗\frac{1}{\sqrt{2}}(|0\rangle +{\left|1\rangle \right)}_{25}. \end{array}$$

Next, we apply 24 CNOT gates, where the 25th qubit serves as the control qubit and the remaining 24 qubits serve as the target qubits. Each CNOT gate entangles the 25th qubit with the corresponding target qubit. After these operations, the final quantum state becomes(4)$$\begin{array}{c} |G_{2}\rangle_{1,2,\cdots ,25}=\frac{1}{\sqrt{2}}(|00\cdots 0\rangle +{\left|11\cdots 1\rangle \right)}_{1,2,\cdots ,25}. \end{array}$$

Subsequently, we proceed with a series of transformations. First, particle 1 passes through a Hadamard gate, followed by a CNOT operation with particle 1 as the control and particle 5 as the target. Then, particle 2 undergoes a Hadamard gate, and a CNOT operation is performed with particle 1 as the control and particle 6 as the target. Particle 3 is next processed with a Hadamard gate, followed by a CNOT operation with particle 3 as the control and particle 7 as the target. Another CNOT operation is applied with particle 3 as the control and particle 4 as the target. Lastly, particle 4 is subjected to a Hadamard gate, and a CNOT operation is performed with particle 4 as the control and particle 8 as the target.

Similarly, we apply the same set of transformations to particles (9,10,⋯,16) and (17,18,⋯,24). After all these steps, the 25-particle maximally entangled channel, as shown in Equation (Equation 1), is fully constructed.

This construction process provides us with a general method for preparing multi-particle entangled states. Using this method, we can construct a maximally entangled state for any number of particles of the form (8n+1).

### 2.2. Four-Party DDC Controlled Quantum Teleportation Protocol

In this subsection, we provide a detailed description of the four-party DDC controlled QT protocol, using the 25-particle entangled state constructed in Section 2.1 as the quantum channel. As illustrated in Figure 1, The protocol involves four participants: Alice, Bob, and Charlie as communicators, and David as the supervisor. Alice intends to send the unknown two-particle state $|\epsilon_{1}\rangle_{A_{1}{\hat{A}}_{1}}$ to Bob and $|\epsilon_{2}\rangle_{A_{2}{\hat{A}}_{2}}$ to Charlie. Bob aims to send $|\omega_{1}\rangle_{B_{1}{\hat{B}}_{1}}$ to Charlie and $|\omega_{2}\rangle_{B_{2}{\hat{B}}_{2}}$ to Alice. Meanwhile, Charlie plans to teleport $|\lambda_{1}\rangle_{C_{1}{\hat{C}}_{2}}$ to Alice and $|\lambda_{2}\rangle_{C_{2}{\hat{C}}_{2}}$ to Bob, all under the supervision of David. These arbitrary unknown two-particle states can be represented as $|\epsilon_{k}\rangle_{X_{k}{\hat{X}}_{k}}=\sum_{l=1}^{4}\mu_{kl}{|b_{l}\rangle }_{X_{k}{\hat{X}}_{k}}$, where k∈{1,2} corresponds to the different quantum states transmitted, and X∈{A,B,C} represents Alice, Bob, or Charlie as the sender. Here, $\mu_{kl}$ are the complex coefficients for each quantum state and $|b_{l}\rangle$ is the two-particle computational basis, with l∈{1,2,3,4} corresponding to |00〉,|01〉,|10〉,|11〉. The coefficients $\mu_{kl}$, $\nu_{kl}$, and $\gamma_{kl}$(for k∈{1,2}, l∈{1,2,3,4}) satisfy the normalization conditions $\sum_{l=1}^{4}{\left|\mu_{kl}\right|}^{2}=1$, $\sum_{l=1}^{4}{\left|\nu_{kl}\right|}^{2}=1$, and $\sum_{l=1}^{4}{\left|\gamma_{kl}\right|}^{2}=1$, ensuring the proper normalization of the quantum states involved. This generalized expression unifies the representation of all the quantum states used in the protocol, eliminating redundancy while keeping the clarity of the different states exchanged between Alice, Bob, and Charlie under the supervisor’s control.

In the preparation phase, the supervisor David needs to prepare a 25-particle maximally entangled state as depicted in Equation (Equation 1), then should retain particle 25 for himself and allocate particles (1,2,3,4,15,16,21,22), (5,6,9,10,11,12,23,24) and (7,8,13,14,17,18,19,20) to Alice, Bob, and Charlie, respectively. This way, David, Alice, Bob, and Charlie share an entangled quantum system, each participant holding a distinct set of particles. The entangled state |G〉 that describes this system is written as a superposition of two terms. Each term consists of the product of three maximally entangled 8-particle states, one for each of Alice, Bob, and Charlie. In the first term, David’s qubit is in the state |0〉, while in the second term, David’s qubit is in the state |1〉. Thus, the entire initial quantum system, combining the individual states $|\epsilon_{1}\rangle$, $|\epsilon_{2}\rangle$, $|\omega_{1}\rangle$, $|\omega_{2}\rangle$, $|\lambda_{1}\rangle$, and $|\lambda_{2}\rangle$, is described by the following expression:(5)$$\begin{array}{c} \left|T\rangle =\right|\epsilon_{1}\rangle_{A_{1}{\hat{A}}_{1}}|\epsilon_{2}\rangle_{A_{2}{\hat{A}}_{2}}|\omega_{1}\rangle_{B_{1}{\hat{B}}_{1}}|\omega_{2}\rangle_{B_{2}{\hat{B}}_{2}}|\lambda_{1}\rangle_{C_{1}{\hat{C}}_{1}}|\lambda_{2}\rangle_{C_{2}{\hat{C}}_{2}}{|G\rangle }_{ABCD}. \end{array}$$
where the bold letters A, B, and C represent the sets of particles $A_{1}^{\prime }A_{2}^{\prime }\cdots A_{8}^{\prime }$, $B_{1}^{\prime }B_{2}^{\prime }\cdots B_{8}^{\prime }$, and $C_{1}^{\prime }C_{2}^{\prime }\cdots C_{8}^{\prime }$, respectively.

In order to fulfil DDC controlled QT, our protocol sequentially executes the following steps, as illustrated in Figure 1.

**Step 1**: Alice, Bob, and Charlie each perform measurements on their respective particle pairs using a Bell-state measurement basis, resulting in the evolution of a complex multipartite quantum entangled state. Alice measures the particle pairs $\left(A_{1},A_{1}^{\prime }\right)$, $\left({\hat{A}}_{1},A_{2}^{\prime }\right)$, $\left(A_{2},A_{3}^{\prime }\right)$, and $\left({\hat{A}}_{2},A_{4}^{\prime }\right)$ with the Bell-state measurement basis $|B_{st}\rangle =\frac{1}{\sqrt{2}}[|0,s\rangle +{\left(-1\right)}^{t}|1,1⊕s\rangle ],$ where s,t∈{0,1}, and ⊕ denotes addition modulo 2. Alice’s measurement results correspond to a superposition of four Bell states, and these measurement outcomes interact with the system’s entanglement, affecting the final state of the system through the complex coefficients $x_{ij}$. After Alice’s measurement, the particle pairs she holds will transform into the following entangled states: $|B_{fg}\rangle_{A_{1}A_{1}^{\prime }}|B_{ht}\rangle_{{\hat{A}}_{1}A_{2}^{\prime }}|B_{uv}\rangle_{A_{2}A_{3}^{\prime }}{|B_{pq}\rangle }_{{\hat{A}}_{2}A_{4}^{\prime }}$. Each Bell state corresponds to a measurement outcome |f,h〉, |f,1⊕h〉, |1⊕f,h〉, and |1⊕f,1⊕h〉. These outcomes, through the complex coefficients $x_{ij}$, become entangled with Bob and Charlie’s measurement results, influencing the overall system’s state.

Bob performs Bell-state measurements on the particle pairs $\left(B_{1},B_{1}^{\prime }\right)$, $\left({\hat{B}}_{1},B_{2}^{\prime }\right)$, $\left(B_{2},B_{3}^{\prime }\right)$, and $\left({\hat{B}}_{2},B_{4}^{\prime }\right)$, the Bell-state basis Bob uses is the same as Alice’s, given by $|B_{st}\rangle =\frac{1}{\sqrt{2}}[|0,s\rangle +{\left(-1\right)}^{t}\left|1,1⊕s\rangle \right]$. Bob’s measurement outcomes depend on Alice’s measurements, so the Bell states Bob measures will be entangled with Alice’s results. Bob’s measurement results will form new particle pair entangled states, which are expressed as $|B_{f^{\prime }g^{\prime }}\rangle_{B_{1}B_{1}^{\prime }}|B_{h^{\prime }t^{\prime }}\rangle_{{\hat{B}}_{1}B_{2}^{\prime }}|B_{u^{\prime }v^{\prime }}\rangle_{B_{2}B_{3}^{\prime }}{|B_{p^{\prime }q^{\prime }}\rangle }_{{\hat{B}}_{2}B_{4}^{\prime }}.$ These measurement outcomes will be entangled with Alice’s and Charlie’s results through the complex coefficients $y_{ij}$, thereby affecting the overall quantum state of the system.

Charlie also measures his four particle pairs $\left(C_{1},C_{1}^{\prime }\right)$, $\left({\hat{C}}_{1},C_{2}^{\prime }\right)$, $\left(C_{2},C_{3}^{\prime }\right)$, and $\left({\hat{C}}_{2},C_{4}^{\prime }\right)$, using the same Bell-state basis as Alice and Bob. Charlie’s measurement outcomes will affect the final quantum state of the system through complex coefficients $z_{ij}$, and these outcomes will become entangled with the results from Alice and Bob. Charlie’s measurement results are expressed as $|B_{f^{\prime \prime }g^{\prime \prime }}\rangle_{C_{1}C_{1}^{\prime }}|B_{h^{\prime \prime }t^{\prime \prime }}\rangle_{{\hat{C}}_{1}C_{2}^{\prime }}|B_{u^{\prime \prime }v^{\prime \prime }}\rangle_{C_{2}C_{3}^{\prime }}{|B_{p^{\prime \prime }q^{\prime \prime }}\rangle }_{{\hat{C}}_{2}C_{4}^{\prime }}$. Through the complex coefficients $z_{ij}$, these measurement outcomes entangle with Alice’s and Bob’s results, ultimately influencing the overall quantum state of the system.

Through Alice, Bob, and Charlie’s Bell-state measurements, the initial quantum state of the system evolves into a complex tensor product form, consisting of multiple Bell-state measurement outcomes and the entanglement generated by these measurements. The measurement results of each participant become entangled with the results of the others, ultimately forming a complex multipartite quantum entangled state |T〉, with the explicit form given in Equation (A4) of Appendix A.

Generally, Alice transfers the measurement results $|B_{fg}\rangle_{A_{1}A_{1}^{\prime }}{|B_{ht}\rangle }_{{\hat{A}}_{1}A_{2}^{\prime }}$ and $|B_{uv}\rangle_{A_{2}A_{3}^{\prime }}{|B_{pq}\rangle }_{{\hat{A}}_{2}A_{4}^{\prime }}$ to Bob and Charlie, respectively. Bob notifies Charlie and Alice of his measurement results $|B_{m^{\prime }n^{\prime }}\rangle_{B_{1}B_{1}^{\prime }}{|B_{r^{\prime }s^{\prime }}\rangle }_{{\hat{B}}_{1}B_{2}^{\prime }}$ and $|B_{i^{\prime }j^{\prime }}\rangle_{B_{2}B_{3}^{\prime }}$$|B_{l^{\prime }k^{\prime }}\rangle_{{\hat{B}}_{2}B_{4}^{\prime }}$, respectively. At the same time, Charlie announces his measurement outcomes A and B to $|B_{m^{\prime \prime }n^{\prime \prime }}\rangle_{C_{1}C_{1}^{\prime }}$$|B_{r^{\prime \prime }s^{\prime \prime }}\rangle_{{\hat{C}}_{1}C_{2}^{\prime }}$ and $|B_{i^{\prime \prime }j^{\prime \prime }}\rangle_{C_{2}C_{3}^{\prime }}{|B_{l^{\prime \prime }k^{\prime \prime }}\rangle }_{{\hat{C}}_{2}C_{4}^{\prime }}$ to Alice and Bob separately. Then, the measured quantum state $|T^{\prime }\rangle$ can be expressed in a concise and structured form, with its explicit expression provided in Equation (A5) of Appendix A.

**Step 2**: If the supervisor, David, agrees to communicate, then he should perform a single-particle Von Neumann measurement on particle *D* using the computational basis {|0〉,|1〉}. Then, he sends a 1-bit classical message *w* (w=0,1) to Alice, Bob, and Charlie over three distinct channels (David–Alice, David–Bob, and David–Charlie), corresponding to the measurement result ${|w\rangle }_{D}$.

**Step 3**: Upon receiving the measurement results from the other participants, Alice, Bob, and Charlie are required to apply the appropriate unitary operations according to the results of the measurements in order to reconstruct the original quantum state. The unitary operations involved in this process can be represented by the following general expression:(6)$$\begin{array}{c} U^{\left(X\right)}=\left(1-w\right)\left[U^{\left(X\right)}⊗{\hat{U}}^{\left(X\right)}\right]+w\left[U^{\prime (X)}⊗{\hat{U}}^{\prime (X)}\right], \end{array}$$
where $X\in \{A^{\prime },B^{\prime },C^{\prime }\}$ represents the different participants Alice, Bob, and Charlie, and its specific form is provided in Equation (A6) of Appendix A. $U^{\left(X\right)}$ and ${\hat{U}}^{\left(X\right)}$ are the first set of unitary operations corresponding to participant *X*, and the specific expressions can be found in Equation (A6) of Appendix A. $U^{\prime (X)}$ and ${\hat{U}}^{\prime (X)}$ are the second set of unitary operations corresponding to participant *X*, and the specific expressions can also be found in Equation (A7) of Appendix A. *w* is a weighting factor that adjusts the contribution of the first and second sets of operations based on the measurement results received by each participant. These unitary operations are applied to the particle groups $\left(A_{5}^{\prime },A_{6}^{\prime },A_{7}^{\prime },A_{8}^{\prime }\right)$, $\left(B_{5}^{\prime },B_{6}^{\prime },B_{7}^{\prime },B_{8}^{\prime }\right)$, and $\left(C_{5}^{\prime },C_{6}^{\prime },C_{7}^{\prime },C_{8}^{\prime }\right)$, enabling each group to recover their intended original states. In other words,(7)$$\begin{array}{ccc} \; & U^{\left(A\right)}U^{\left(B\right)}U^{\left(C\right)}{(}_{D}\langle t|T^{\prime }\rangle ) & \\ \; & =(x_{11}|00\rangle +x_{12}|01\rangle +x_{13}|10\rangle +x_{14}{\left|11\rangle \right)}_{B_{5}^{\prime }B_{6}^{\prime }} & \\ \; & \;\;⊗(z_{21}|00\rangle +z_{22}|01\rangle +z_{23}|10\rangle +z_{24}{\left|11\rangle \right)}_{B_{7}^{\prime }B_{8}^{\prime }} & \\ \; & \;\;⊗(y_{11}|00\rangle +y_{12}|01\rangle +y_{13}|10\rangle +y_{14}{\left|11\rangle \right)}_{C_{5}^{\prime }C_{6}^{\prime }} & \\ \; & \;\;⊗(x_{21}|00\rangle +x_{22}|01\rangle +x_{23}|10\rangle +x_{24}{\left|11\rangle \right)}_{C_{7}^{\prime }C_{8}^{\prime }} & \\ \; & \;\;⊗(z_{11}|00\rangle +z_{12}|01\rangle +z_{13}|10\rangle +z_{14}{\left|11\rangle \right)}_{A_{5}^{\prime }A_{6}^{\prime }} & \\ \; & \;\;⊗(y_{21}|00\rangle +y_{22}|01\rangle +y_{23}|10\rangle +y_{24}{\left|11\rangle \right)}_{A_{7}^{\prime }A_{8}^{\prime }}. & \end{array}$$

From Equation (Equation 7), it can be observed that Alice’s state $|\epsilon_{1}\rangle_{A_{1}^{\prime }{\hat{A}}_{1}}$ has been teleported to Bob, and $|\epsilon_{2}\rangle_{A_{2}^{\prime }{\hat{A}}_{2}}$ to Charlie. Bob’s state $|\omega_{1}\rangle_{B_{1}{\hat{B}}_{1}}$ has been transferred to Charlie, while $|\omega_{2}\rangle_{B_{2}{\hat{B}}_{2}}$ has been sent to Alice. Meanwhile, Charlie’s state $|\lambda_{1}\rangle_{C_{1}{\hat{C}}_{1}}$ has been transmitted to Alice, and $|\lambda_{2}\rangle_{C_{2}{\hat{C}}_{2}}$ to Bob.

Additionally, according to Equation (A4) in Appendix A, our scheme has 12 Bell-state measurements and 1 single-particle von Neumann measurement, yielding a total of $4^{4}\times 4^{4}\times 4^{4}\times 2=33,554,432$ possible measurement outcomes. For each outcome, the unitary transformations in Equation (Equation 6) are applied to correctly reconstruct the desired states. As a result, our scheme achieves a success probability of 100%.

### 2.3. Generalized DDC Controlled Quantum Teleportation Scheme

In this subsection, we extend our four-party scheme of DDC controlled QT to the scene with *n* (n>3) communicators in this subsection. Assume the *n* communicators can be denoted as $\left\{N_{1},N_{2},\cdots ,N_{n}\right\}$, and they form a closed ring. All communicators initially share an (8n+1)-particle maximally entangled state together with the supervisor, Tom. This state can be written as(8)$$\begin{array}{ccc} {|G\rangle }_{N_{1}N_{2}\cdots N_{n}} & =\frac{1}{\sqrt{2}}[\overset{n}{\underset{j=1}{⨂}}|\varphi_{1}^{8}\rangle_{N_{j}^{1}N_{j}^{2}N_{j}^{3}N_{j}^{4}N_{j+1}^{5}N_{j+1}^{6}N_{j-1}^{7}N_{j-1}^{8}}{|0\rangle }_{T} & \\ \; & +\overset{n}{\underset{j=1}{⨂}}|\varphi_{2}^{8}\rangle_{N_{j}^{1}N_{j}^{2}N_{j}^{3}N_{j}^{4}N_{j+1}^{5}N_{j+1}^{6}N_{j-1}^{7}N_{j-1}^{8}}{|1\rangle }_{T}], \end{array}$$
where the communicator $N_{j}$ has eight particles $\left(N_{j}^{1},N_{j}^{2},\cdots ,N_{j}^{8}\right)$ for any j∈{1,2,⋯,n}. Additionally, the particle *T* belongs to the supervisor Tom. For j=n, we impose $N_{n+1}^{5}=N_{1}^{5}$ and $N_{n+1}^{6}=N_{1}^{6}$, and for j=1, we impose $N_{0}^{7}=N_{n}^{7}$ and $N_{0}^{8}=N_{n}^{8}$. The symbol ⨂ represents the tensor product, while $N_{1}$, $N_{2}$, and $N_{n}$ denote the groups of particles $N_{1}^{1}N_{1}^{2}\cdots N_{1}^{8}$, $N_{2}^{1}N_{2}^{2}\cdots N_{2}^{8}$, and $N_{n}^{1}N_{n}^{2}\cdots N_{n}^{8}$, respectively. The entangled states $|\varphi_{1}^{8}\rangle$ and $|\varphi_{2}^{8}\rangle$ are defined in Equations (A1) and (A2) of Appendix A, respectively.

Consider that $N_{1}$ intends to teleport two arbitrary unknown two-particle states, $|\epsilon^{\prime }\rangle_{N_{1}^{\prime }}$ and $|\epsilon^{2}\rangle_{N_{1}^{\prime \prime }}$, to $N_{2}$ and $N_{n}$, respectively. Each participant $N_{j}$ (j∈{2,3,⋯,n−1}) aims to send two arbitrary unknown two-particle states, $|\epsilon^{\prime }\rangle_{N_{j}^{\prime }}$ and $|\epsilon^{2}\rangle_{N_{j}^{\prime \prime }}$, to $N_{j+1}$ and $N_{j-1}$, respectively. Finally, $N_{n}$ intends to transmit the state $|\epsilon^{\prime }\rangle_{N_{n}^{\prime }}$ to $N_{1}$ and $|\epsilon^{2}\rangle_{N_{n}^{\prime \prime }}$ to $N_{n-1}$, with the entire process being supervised and controlled by Tom. The relation among these n+1 participants is shown in Figure 3.

The above 2n arbitrary unknown two-particle states to be teleported can be expressed as$$|\epsilon^{\prime }\rangle_{N_{j}^{\prime }}=a_{11}^{\left(N_{j}^{\prime }\right)}|00\rangle +a_{12}^{\left(N_{j}^{\prime }\right)}|01\rangle +a_{13}^{\left(N_{j}^{\prime }\right)}|10\rangle +a_{14}^{\left(N_{j}^{\prime }\right)}|11\rangle$$
where complex coefficients $a_{k1}^{\left(N_{j}^{\prime }\right)},a_{k2}^{\left(N_{j}^{\prime }\right)},a_{k3}^{\left(N_{j}^{\prime }\right)}$, and $a_{k4}^{\left(N_{j}^{\prime }\right)}$ satisfy the normalization condition $|a_{k1}^{\left(N_{j}^{\prime }\right)}{|}^{2}+|a_{k2}^{\left(N_{j}^{\prime }\right)}{|}^{2}+|a_{k3}^{\left(N_{j}^{\prime }\right)}{|}^{2}+{\left|a_{k4}^{\left(N_{j}^{\prime }\right)}\right|}^{2}=1$ (k=1,2). The second unknown two-particle state, $|\epsilon^{2}\rangle_{N_{j}^{\prime }}$, can be expressed in terms of the same form as $|\epsilon^{\prime }\rangle_{N_{j}^{\prime }}$, with its coefficients denoted as $a_{21}^{\left(N_{j}^{\prime }\right)},a_{22}^{\left(N_{j}^{\prime }\right)},a_{23}^{\left(N_{j}^{\prime }\right)},a_{24}^{\left(N_{j}^{\prime }\right)}$, which similarly satisfy the normalization condition. Thus, $|\epsilon^{2}\rangle_{N_{j}^{\prime }}$ can be written by replacing $a_{1k}^{\left(N_{j}^{\prime }\right)}$ in $|\epsilon^{\prime }\rangle_{N_{j}^{\prime }}$ with $a_{2k}^{\left(N_{j}^{\prime }\right)}\left(k=1,2,3,4\right)$. The complete initial system state can be represented as follows:(9)$$\begin{array}{c} |W\rangle ={⊗}_{j=1}^{n}|\epsilon^{\prime }\rangle_{N_{j}^{\prime }}{⊗}_{j=1}^{n}|\epsilon^{2}\rangle_{N_{j}^{\prime \prime }}⊗{|G\rangle }_{N_{1}N_{2}\cdots N_{n}}. \end{array}$$

In order to complete DDC controlled QT of arbitrary unknown two-particle states among n+1 participants, each communicator needs to execute four Bell-state measurements and then inform the two adjacent communicators of the measurement results. After that, the supervisor Tom makes a single-particle projective measurement on his particle *T* in the basis {|0〉,|1〉}, and relays his measurement outcome to all communicators. According to the outcomes from the two adjacent communicators and the controller, each communicator is able to successfully reconstruct the target two-particle states. For simplicity, we will omit the detailed discussion on the connections between measurement outcomes, collapsed states, and their corresponding recovery operations. Since this extended scheme employs procedures and operations similar to those in the earlier scheme with three communicators and one supervisor, the success probability of our proposed scheme remains 1.

## 3. DDC Controlled RSP of Arbitrary Two-Particle States

Based on the quantum channel [1] we constructed, this subsection proposes a four-party DDC controlled RSP scheme, which can be used for the preparation of arbitrary two-particle states and further extended to scenarios involving n+1 (n>3) participants. As the quantum circuit diagram illustrating the relationships among the four participants is similar to that in Section 2, it is not repeated here.

### 3.1. Four-Party DDC Controlled RSP Scheme

Assume that Alice wishes to help Bob prepare an arbitrary two-particle state $|\phi_{1}\rangle$ remotely and assist Charlie in preparing $|\phi_{2}\rangle$; Bob intends to aid Charlie remotely with the preparation of $|\psi_{1}\rangle$ and also help Alice prepare $|\psi_{2}\rangle$. At the same time, Charlie plans to help Alice prepare an arbitrary two-particle state $|\chi_{1}\rangle$ remotely and assist Bob in preparing $|\chi_{2}\rangle$, all under the supervision of David. The six arbitrary two-particle states to be prepared can be written as follows:(10)$$\begin{array}{cc} \; & |\phi_{1}\rangle =\sum_{l=1}^{4}a_{1l}e^{i\theta_{l}}|b_{l}\rangle ,\;|\phi_{2}\rangle =\sum_{l=1}^{4}a_{2l}e^{i\theta_{l}^{\prime }}|b_{l}\rangle ,\\ \; & |\psi_{1}\rangle =\sum_{l=1}^{4}b_{1l}e^{i\alpha_{l}}|b_{l}\rangle ,\;|\psi_{2}\rangle =\sum_{l=1}^{4}b_{2l}e^{i\alpha_{l}^{\prime }}|b_{l}\rangle ,\\ \; & |\chi_{1}\rangle =\sum_{l=1}^{4}c_{1l}e^{i\beta_{l}}|b_{l}\rangle ,\;|\chi_{2}\rangle =\sum_{l=1}^{4}c_{2l}e^{i\beta_{l}^{\prime }}|b_{l}\rangle , \end{array}$$
where real numbers $a_{kl},b_{kl}$, $c_{kl}$, $\theta_{l},\theta_{l}^{\prime },\alpha_{l},\alpha_{l}^{\prime },\beta_{l}$, and $\beta_{l}^{\prime }$ (k∈{1,2}, l∈{1,2,3,4}) satisfy the conditions $\Sigma_{l=1}^{4}{\left|a_{kl}\right|}^{2}=1$, $\Sigma_{l=1}^{4}{\left|b_{kl}\right|}^{2}=1$, $\Sigma_{l=1}^{4}{\left|c_{kl}\right|}^{2}=1$, and $\theta_{l},\theta_{l}^{\prime },\alpha_{l},\alpha_{l}^{\prime },\beta_{l},\beta_{l}^{\prime }\in \left[0,2\pi \right)$. Note that for any k∈{1,2} and l∈{1,2,3,4}, Alice is completely aware of coefficients $a_{kl}$, $\theta_{l}$, and $\theta_{l}^{\prime }$, but Bob and Charlie are not aware of them. Similarly, Bob knows the coefficients $b_{kl}$, $\alpha_{l}$, and $\alpha_{l}^{\prime }$, but Charlie and Alice do not know about them, and the coefficients $c_{kl}$, $\beta_{l}$, and $\beta_{l}^{\prime }$ are known to Charlie but unknown to Alice and Bob.

Similar to Section 2.2, a 25-particle maximally entangled channel is pre-shared among three communicators—Alice, Bob, and Charlie—and the supervisor David.

To achieve the quantum task of four-party DDC-controlled RSP, the following steps must be performed:

**Step 1**: Three communicators, Alice, Bob, and Charlie, introduce three auxiliary particles $({\bar{A}}_{1},{\bar{A}}_{2},{\bar{A}}_{3},$${\bar{A}}_{4})$, $\left({\bar{B}}_{1},{\bar{B}}_{2},{\bar{B}}_{3},{\bar{B}}_{4}\right)$, and $\left({\bar{C}}_{1},{\bar{C}}_{2},{\bar{C}}_{3},{\bar{C}}_{4}\right)$, respectively. In this way, the initial system state of 37 particles can be represented as(11)$$\begin{array}{c} |T\rangle ={|G\rangle }_{ABCD}{|0000\rangle }_{\bar{A}}{|0000\rangle }_{\bar{B}}{|0000\rangle }_{\bar{C}}, \end{array}$$
where the bold letters A, B, and C represent the sets of particles $A_{1}^{\prime }A_{2}^{\prime }\cdots A_{8}^{\prime }$, $B_{1}^{\prime }B_{2}^{\prime }\cdots B_{8}^{\prime }$, and $C_{1}^{\prime }C_{2}^{\prime }\cdots C_{8}^{\prime }$, respectively. Similarly, $\bar{A}$, $\bar{B}$, and $\bar{C}$ represent the sets of particles ${\bar{A}}_{1}{\bar{A}}_{2}{\bar{A}}_{3}{\bar{A}}_{4})$, ${\bar{B}}_{1}{\bar{B}}_{2}{\bar{B}}_{3}{\bar{B}}_{4}$, and ${\bar{C}}_{1}{\bar{C}}_{2}{\bar{C}}_{3}{\bar{C}}_{4}$, respectively.

Then, Alice carries out the CNOT operation $C_{XY}$ on qubit pairs $\left(A_{1}^{\prime },{\bar{A}}_{1}\right)$, $\left(A_{2}^{\prime },{\bar{A}}_{2}\right)$, $\left(A_{3}^{\prime },{\bar{A}}_{3}\right)$, and $\left(A_{4}^{\prime },{\bar{A}}_{4}\right)$, respectively, where $C_{XY}{|uv\rangle }_{XY}={|u\rangle }_{X}{|u⊕v\rangle }_{Y}$. That is, qubits $A_{1}^{\prime },A_{2}^{\prime },A_{3}^{\prime },A_{4}^{\prime }$ serve as controlling qubits and auxiliary qubits ${\bar{A}}_{1},{\bar{A}}_{2},{\bar{A}}_{3},{\bar{A}}_{4}$ function as target qubits. After Bob and Charlie also perform similar operations, the state |T〉 shown in Equation (Equation 11) will change to(12)$$\begin{array}{c} |T^{\prime }\rangle =\frac{1}{\sqrt{2}}\overset{3}{\underset{j=1}{⨂}}|\varphi_{1}^{12}\rangle_{P_{j}}⊗{|0\rangle }_{D}+\frac{1}{\sqrt{2}}\overset{3}{\underset{j=1}{⨂}}|\varphi_{2}^{12}\rangle_{P_{j}}⊗{|1\rangle }_{D}, \end{array}$$
where the $P_{1},P_{2},P_{3}$ are defined as follows: $P_{1}=A_{1}^{\prime }A_{2}^{\prime }{\bar{A}}_{1}{\bar{A}}_{2}A_{3}^{\prime }A_{4}^{\prime }{\bar{A}}_{3}{\bar{A}}_{4}B_{5}^{\prime }B_{6}^{\prime }C_{7}^{\prime }C_{8}^{\prime },$ $P_{2}=B_{1}^{\prime }B_{2}^{\prime }{\bar{B}}_{1}{\bar{B}}_{2}B_{3}^{\prime }B_{4}^{\prime }{\bar{B}}_{3}{\bar{B}}_{4}C_{5}^{\prime }C_{6}^{\prime }A_{7}^{\prime }A_{8}^{\prime },$ and $P_{3}=C_{1}^{\prime }C_{2}^{\prime }{\bar{C}}_{1}{\bar{C}}_{2}C_{3}^{\prime }C_{4}^{\prime }{\bar{C}}_{3}{\bar{C}}_{4}A_{5}^{\prime }A_{6}^{\prime }B_{7}^{\prime }B_{8}^{\prime }.$ The detailed forms of $|\varphi_{1}^{12}\rangle$ and $|\varphi_{2}^{12}\rangle$ are provided in Appendix A as Equations (A8) and (A9), respectively. For brevity, their explicit expressions are deferred to the appendix.

**Step 2**: Alice performs two projective measurements on $\left(A_{1}^{\prime },A_{2}^{\prime }\right)$ and $\left(A_{3}^{\prime },A_{4}^{\prime }\right)$ with the measurement basis $\left\{|\epsilon_{uv}|u,v=0,1\right\}$ and $\left\{|\zeta_{pq}|p,q=0,1\right\}$, respectively. These measurement bases are defined as weighted combinations of the standard two-qubit basis states (|00〉,|01〉,|10〉,|11〉), where the coefficients and phase factors depend on the measurement indices. For the first type of measurement basis, the general expression is(13)$$\begin{array}{ccc} |\epsilon_{uv}\rangle & =a_{1,2u+v+1}|00\rangle +{\left(-1\right)}^{u+v}a_{1,2u+v+1+{\left(-1\right)}^{v}}|01\rangle & \\ \; & +{\left(-1\right)}^{u}a_{1,2u+v+1+2{\left(-1\right)}^{u}}|10\rangle +{\left(-1\right)}^{v}a_{1,4-2u-v}|11\rangle , & \end{array}$$
where u,v∈{0,1}, and the coefficients $a_{1,k}$ are participant-specific weights that determine the contribution of each basis state. Similarly, the second type of measurement basis, denoted as $|\zeta_{pq}\rangle$, has a structure analogous to $|\epsilon_{uv}\rangle$. The differences lie in the replacement of indices u,v with p,q, and the coefficients $a_{1,k}$ are replaced by $a_{2,k}$. Additionally, logical operations such as 1⊕q and p⊕q modify the relative phases of the basis states.

Bob and Charlie use similar measurement bases. For Bob, the first and second types of bases are denoted as $|\epsilon_{u^{\prime }v^{\prime }}^{\prime }\rangle$ and $|\zeta_{p^{\prime }q^{\prime }}^{\prime }\rangle$, following the same structure but with coefficients $b_{1,k}$ and $b_{2,k}$, respectively. Charlie, on the other hand, uses the bases $|\epsilon_{u^{\prime \prime }v^{\prime \prime }}^{\prime \prime }\rangle$ and $|\zeta_{p^{\prime \prime }q^{\prime \prime }}^{\prime \prime }\rangle$, where the coefficients are $c_{1,k}$ and $c_{2,k}$. The indices $u,v,u^{\prime },v^{\prime },u^{\prime \prime },v^{\prime \prime },p,q,p^{\prime },q^{\prime },p^{\prime \prime },q^{\prime \prime }\in \left\{0,1\right\}$ represent binary measurement outcomes. These indices determine the specific coefficients and phase factors in the measurement bases, encoding the results of the measurements. After completing the measurements, the results are exchanged among the participants. Alice sends her outcomes $|\epsilon_{uv}\rangle$ to Bob and $|\zeta_{pq}\rangle$ to Charlie. Bob shares $|\epsilon_{u^{\prime }v^{\prime }}^{\prime }\rangle$ with Charlie and $|\zeta_{p^{\prime }q^{\prime }}^{\prime }\rangle$ with Alice. Similarly, Charlie transfers $|\epsilon_{u^{\prime \prime }v^{\prime \prime }}^{\prime \prime }\rangle$ to Alice and $|\zeta_{p^{\prime \prime }q^{\prime \prime }}^{\prime \prime }\rangle$ to Bob.

**Step 3**: After receiving the measurement results, Alice measures her particle pairs $\left({\bar{A}}_{1},{\bar{A}}_{2}\right)$ and $\left({\bar{A}}_{3},{\bar{A}}_{4}\right)$ using a feedforward measurement strategy. For $\left({\bar{A}}_{1},{\bar{A}}_{2}\right)$, she constructs the measurement basis $\left\{|\omega_{st}^{\left(uv\right)}\rangle |s,t=0,1\right\}$, where each state is expressed as(14)$$\begin{array}{ccc} |\omega_{st}^{\left(uv\right)}\rangle & =\frac{1}{2}[e^{-i\theta_{2u+v+1}}|00\rangle +{\left(-1\right)}^{s+t}e^{-i\theta_{2u+v+1+{\left(-1\right)}^{v}}}|01\rangle & \\ \; & \;\;+{\left(-1\right)}^{s}e^{-i\theta_{2u+v+1+2{\left(-1\right)}^{u}}}|10\rangle +{\left(-1\right)}^{t}e^{-i\theta_{4-2u-v}}|11\rangle ], \end{array}$$
where u,v∈{0,1} and s,t∈{0,1}. Here, the coefficients $e^{-i\theta_{n}}$ introduce phase adjustments for each basis state, and the terms ${\left(-1\right)}^{s+t}$, ${\left(-1\right)}^{s}$, and ${\left(-1\right)}^{t}$ account for relative phase differences. Similarly, for the second pair of qubits $\left({\bar{A}}_{3},{\bar{A}}_{4}\right)$, she constructs the measurement basis $\left\{|\varsigma_{mr}^{\left(pq\right)}\rangle |m,r=0,1\right\}$, which follows a similar structure to $|\omega_{st}^{\left(uv\right)}\rangle$. Specifically, in this case, the indices u,v and s,t are replaced by p,q and m,r, respectively. The corresponding phase factors $\theta_{n}$ are adjusted accordingly to reflect the changes in indices. This unified structure ensures symmetry and consistency in the measurement bases for different qubit pairs.

Bob and Charlie follow a similar measurement process to Alice. Bob measures his particle pairs $\left({\bar{B}}_{1},{\bar{B}}_{2}\right)$ and $\left({\bar{B}}_{3},{\bar{B}}_{4}\right)$, constructing the bases $\{|{\hat{\omega }}_{s^{\prime }t^{\prime }}^{\left(u^{\prime }v^{\prime }\right)}\rangle \left|s^{\prime },t^{\prime }=0,1\right\}$ and $\{|{\hat{\varsigma }}_{m^{\prime }r^{\prime }}^{\left(p^{\prime }q^{\prime }\right)}\rangle \left|m^{\prime },r^{\prime }=0,1\right\}$, where the structure of the states mirrors Alice’s, but with participant-specific phase parameters $\alpha_{k}$. Similarly, Charlie measures his particle pairs $\left({\bar{C}}_{1},{\bar{C}}_{2}\right)$ and $\left({\bar{C}}_{3},{\bar{C}}_{4}\right)$ using the bases $\{|{\bar{\omega }}_{s^{\prime \prime }t^{\prime \prime }}^{\left(u^{\prime \prime }v^{\prime \prime }\right)}\rangle \left|s^{\prime \prime },t^{\prime \prime }=0,1\right\}$ and $\{|{\bar{\varsigma }}_{m^{\prime \prime }r^{\prime \prime }}^{\left(p^{\prime \prime }q^{\prime \prime }\right)}\rangle \left|m^{\prime \prime },r^{\prime \prime }=0,1\right\}$, where the phase parameters $\beta_{k}$ are specific to Charlie. Subsequently, each correspondent of the three correspondents needs to send his/her outcomes to the other two correspondents, respectively. Alice sends her outcomes $|\omega_{st}^{\left(uv\right)}\rangle$ to Bob and $|\varsigma_{mr}^{\left(pq\right)}\rangle$ to Charlie. Bob shares $|{\hat{\omega }}_{s^{\prime }t^{\prime }}^{\left(u^{\prime }v^{\prime }\right)}\rangle$ with Charlie and $|{\hat{\varsigma }}_{m^{\prime }r^{\prime }}^{\left(p^{\prime }q^{\prime }\right)}\rangle$ with Alice. Similarly, Charlie shares $|{\bar{\omega }}_{s^{\prime \prime }t^{\prime \prime }}^{\left(u^{\prime \prime }v^{\prime \prime }\right)}\rangle$ to Alice and $|{\bar{\varsigma }}_{m^{\prime \prime }r^{\prime \prime }}^{\left(p^{\prime \prime }q^{\prime \prime }\right)}\rangle$ to Bob. This mutual sharing of measurement outcomes ensures that all three participants can synchronize their operations for the subsequent steps in the protocol.

By applying the six sets of measurement bases described above, the quantum state $|T^{\prime }\rangle$ introduced in Equation (Equation 12) can be rewritten in a detailed expanded form. The full mathematical expression of $|T^{\prime }\rangle$ is provided in Appendix A as Equation (A10).

**Step 4**: If supervisor David agrees to help the three communicators, he performs a single-particle von Neumann measurement on his particle *D* in the {|0〉,|1〉} basis.He then informs the communicators of his measurement result, denoted as ${|d\rangle }_{D}$ (d=0,1).

**Step 5**: After hearing the classic messages corresponding to the measurement results, each of the three communicators needs to perform an appropriate unitary transformation to restore their respective target states. In detail, after receiving Charlie’s measurement result $|\epsilon_{u^{\prime \prime }v^{\prime \prime }}^{\prime \prime }\rangle_{C_{1}^{\prime }C_{2}^{\prime }}{|{\bar{\omega }}_{s^{\prime \prime }t^{\prime \prime }}^{\left(u^{\prime \prime }v^{\prime \prime }\right)}\rangle }_{{\bar{C}}_{1}{\bar{C}}_{2}}$, Bob’s measurement result $|\zeta_{p^{\prime }q^{\prime }}^{\prime }\rangle_{B_{3}^{\prime }B_{4}^{\prime }}{|{\hat{\varsigma }}_{m^{\prime }r^{\prime }}^{\left(p^{\prime }q^{\prime }\right)}\rangle }_{{\bar{B}}_{3}{\bar{B}}_{4}}$ and David’s measurement result ${|d\rangle }_{D}$, Alice selects the unitary operation(15)$$\begin{array}{ccc} \; & \left(1-d\right)\left[\sigma_{A_{5}^{\prime }}^{\left(s^{\prime \prime },s^{\prime \prime }\right)}⊗\sigma_{A_{6}^{\prime }}^{\left(s^{\prime \prime }⊕t^{\prime \prime },s^{\prime \prime }⊕t^{\prime \prime }\right)}\right]\left[\sigma_{A_{5}^{\prime }}^{\left(u^{\prime \prime },0\right)}⊗\sigma_{A_{6}^{\prime }}^{\left(u^{\prime \prime }⊕v^{\prime \prime },u^{\prime \prime }\right)}\right] & \\ \; & ⊗\left[\sigma_{A_{7}^{\prime }}^{\left(m^{\prime },m^{\prime }\right)}⊕\sigma_{A_{8}^{\prime }}^{\left(1⊕m^{\prime }⊕r^{\prime },1⊕m^{\prime }⊕r^{\prime }\right)}\right]\left[\sigma_{A_{7}^{\prime }}^{\left(p^{\prime }⊕q^{\prime },q^{\prime }\right)}⊕\sigma_{A_{8}^{\prime }}^{\left(\left(p^{\prime }⊕q^{\prime }\right)q^{\prime },p^{\prime }q^{\prime }\right)}\right] & \\ \; & +d\left[\sigma_{A_{5}^{\prime }}^{\left(1⊕s^{\prime \prime },s^{\prime \prime }\right)}⊗\sigma_{A_{6}^{\prime }}^{\left(1⊕s^{\prime \prime }⊕t^{\prime \prime },s^{\prime \prime }⊕t^{\prime \prime }\right)}\right]\left[\sigma_{A_{5}^{\prime }}^{\left(u^{\prime \prime },0\right)}⊗\sigma_{A_{6}^{\prime }}^{\left(u^{\prime \prime }⊕v^{\prime \prime },u^{\prime \prime }\right)}\right] & \\ \; & ⊗\left[\sigma_{A_{7}^{\prime }}^{m^{\prime }⊕1,m^{\prime })}⊗\sigma_{A_{8}^{\prime }}^{\left(m^{\prime }⊕r^{\prime },1⊕m^{\prime }⊕r^{\prime }\right)}\right]\left[\sigma_{A_{7}^{\prime }}^{\left(p^{\prime }⊕q^{\prime },q^{\prime }\right)}⊗\sigma_{A_{8}^{\prime }}^{\left(q^{\prime },0\right)}\right] & \end{array}$$
to reconstruct the original quantum states $|\chi_{1}\rangle_{A_{5}^{\prime }A_{6}^{\prime }}⊗{|\psi_{2}\rangle }_{A_{7}^{\prime }A_{8}^{\prime }}$, where $\sigma^{\left(i,j\right)}={|0\rangle \langle i⊕j|+\left(-1\right)}^{i}|1\rangle \langle 1⊕i⊕j|\;\;\left(i,j=0,1\right)$ are Pauli gate operations. After receiving the classical measurement results, Bob and Charlie perform the appropriate unitary transformations to recover their respective target quantum states. The transformation structure for Bob and Charlie is identical to that of Alice. Specifically Bob’s operation can be derived by replacing the parameters $A_{5}^{\prime },A_{6}^{\prime },A_{7}^{\prime },A_{8}^{\prime }$ in Alice’s transformation formula with $B_{5}^{\prime },B_{6}^{\prime },B_{7}^{\prime },B_{8}^{\prime }$, and substituting the measurement parameters $s^{\prime \prime },t^{\prime \prime },u^{\prime \prime },v^{\prime \prime },p^{\prime },q^{\prime },m^{\prime },r^{\prime }$ with $s,t,u,v,p^{\prime \prime },q^{\prime \prime },m^{\prime \prime },r^{\prime \prime }$. Similarly, Charlie’s operation can be obtained by replacing $A_{5}^{\prime },A_{6}^{\prime },A_{7}^{\prime },A_{8}^{\prime }$ with $C_{5}^{\prime },C_{6}^{\prime },C_{7}^{\prime },C_{8}^{\prime }$, and substituting $s^{\prime \prime },t^{\prime \prime },u^{\prime \prime },v^{\prime \prime },p^{\prime },q^{\prime },m^{\prime },r^{\prime }$ with $s^{\prime },t^{\prime },u^{\prime },v^{\prime },p,q,m,r$. Therefore, it is unnecessary to explicitly write out Bob’s and Charlie’s formulas; their operations can be deduced directly from Alice’s formula by applying the appropriate substitutions.

According to the above derivation, it is evident that the four-party DDC controlled RSP of arbitrary two-particle states is always achievable, ensuring that the total success probability of our scheme is 100%.

### 3.2. Generalized DDC Controlled RSP Scheme

To address the diverse requirements of future quantum communication networks, it is essential to generalize the four-party DDC controlled RSP scheme for arbitrary two-particle states to accommodate *n* (n>3) communication parties. Consider *n* correspondents $N_{1},N_{2},\cdots ,N_{n}$, who pre-share an (8n+1)-particle maximally entangled state with the supervisor, Tom, as described in Equation (Equation 11). Specifically, each eight-particle group $\left(N_{j}^{1},N_{j}^{2},\cdots ,N_{j}^{8}\right)$ is assigned to the correspondent $N_{j}$ (j=1,2,⋯,n), while the single particle *T* is held by Tom. In this scheme, $N_{1}$ assists $N_{2}$ in remotely preparing an arbitrary two-particle state $|\varphi_{N_{1}}^{1}\rangle$ and helps $N_{n}$ prepare $|\varphi_{N_{1}}^{2}\rangle$. Similarly, $N_{j}$ (j=2,3,⋯,n−1) aids $N_{j+1}$ in preparing the state $|\varphi_{N_{j}}^{1}\rangle$ and $N_{j-1}$ in preparing $|\varphi_{N_{j}}^{2}\rangle$. Meanwhile, $N_{n}$ assists $N_{1}$ in preparing $|\varphi_{N_{n}}^{1}\rangle$ and helps $N_{n-1}$ prepare $|\varphi_{N_{n}}^{2}\rangle$, all under the supervision of Tom. The 2n arbitrary two-particle states can be mathematically described using the following general form. The first state, $|\varphi_{N_{j}}^{1}\rangle$, for each correspondent $N_{j}$, is expressed as(16)$$\begin{array}{c} |\varphi_{N_{j}}^{1}\rangle =a_{11}^{\left(j\right)}e^{i\theta_{1}^{j}}|00\rangle +a_{12}^{\left(j\right)}e^{i\theta_{2}^{j}}|01\rangle +a_{13}^{\left(j\right)}e^{i\theta_{3}^{j}}|10\rangle +a_{14}^{\left(j\right)}e^{i\theta_{4}^{j}}|11\rangle , \end{array}$$
where the coefficients $a_{1k}^{\left(j\right)}$ (k=1,2,3,4) satisfy the normalization condition $\sum_{k=1}^{4}{\left|a_{1k}^{\left(j\right)}\right|}^{2}=1$, and the phase parameters $\theta_{k}^{j}\in \left[0,2\pi \right)$. Similarly, the second state, $|\varphi_{N_{j}}^{2}\rangle$, is represented with the same structure, substituting $a_{1k}^{\left(j\right)}$ with $a_{2k}^{\left(j\right)}$ and $\theta_{k}^{j}$ with ${\hat{\theta }}_{k}^{j}$.

In order to complete the quantum task, each correspondent $N_{w}$ (w=1,2,⋯,n) introduces four auxiliary particles $\left({\hat{N}}_{w}^{1},{\hat{N}}_{w}^{2},{\hat{N}}_{w}^{3},{\hat{N}}_{w}^{4}\right)$, initialized in the state ${|0000\rangle }_{{\hat{N}}_{w}^{1}{\hat{N}}_{w}^{2}{\hat{N}}_{w}^{3}{\hat{N}}_{w}^{4}}$. Subsequently, four CNOT gate operations are performed on the particle pairs $\left(N_{w}^{1},{\hat{N}}_{w}^{1}\right)$, $\left(N_{w}^{2},{\hat{N}}_{w}^{2}\right)$, $\left(N_{w}^{3},{\hat{N}}_{w}^{3}\right)$ and $\left(N_{w}^{4},{\hat{N}}_{w}^{4}\right)$, where the particles $N_{w}^{1},N_{w}^{2},N_{w}^{3},N_{w}^{4}$ act as control particles, and the auxiliary particles ${\hat{N}}_{w}^{1},{\hat{N}}_{w}^{2},{\hat{N}}_{w}^{3},{\hat{N}}_{w}^{4}$ serve as targets. After these operations, the entangled channel in Equation (Equation 8) transforms into(17)$$\begin{array}{ccc} |H\rangle = & \frac{1}{\sqrt{2}}[\overset{n}{\underset{j=1}{⨂}}|\varphi_{1}^{12}\rangle_{N_{j}^{1}N_{j}^{2}{\hat{N}}_{j}^{1}{\hat{N}}_{j}^{2}N_{j}^{3}N_{j}^{4}{\hat{N}}_{j}^{3}{\hat{N}}_{j}^{4}N_{j+1}^{5}N_{j+1}^{6}N_{j-1}^{7}N_{j-1}^{8}}{|0\rangle }_{T} & \\ \; & +\overset{n}{\underset{j=1}{⨂}}|\varphi_{2}^{12}\rangle_{N_{j}^{1}N_{j}^{2}{\hat{N}}_{j}^{1}{\hat{N}}_{j}^{2}N_{j}^{3}N_{j}^{4}{\hat{N}}_{j}^{3}{\hat{N}}_{j}^{4}N_{j+1}^{5}N_{j+1}^{6}N_{j-1}^{7}N_{j-1}^{8}}{|1\rangle }_{T}], & \end{array}$$
where $|\varphi_{1}^{12}\rangle$ and $|\varphi_{2}^{12}\rangle$ are the same as Equations (A8) and (A9) in Appendix A, respectively. To ensure the closed-loop structure of the indices, for j=n, we define $N_{n+1}^{5}=N_{1}^{5}$ and $N_{n+1}^{6}=N_{1}^{6}$, while for j=1, we define $N_{j-1}^{7}=N_{n}^{7}$ and $N_{j-1}^{8}=N_{n}^{8}$. The symbol ⨂ represents the tensor product, and the particle *T* belongs to the supervisor. Second, each correspondent $N_{w}$ implements four projective measurements on particle pairs $\left(N_{w}^{1},{\hat{N}}_{w}^{1}\right)$, $\left(N_{w}^{2},{\hat{N}}_{w}^{2}\right)$, $\left(N_{w}^{3},{\hat{N}}_{w}^{3}\right)$, and $\left(N_{w}^{4},{\hat{N}}_{w}^{4}\right)$. Specifically, for the first particle pair $\left(N_{w}^{1},{\hat{N}}_{w}^{1}\right)$, the measurement basis is defined as(18)$$\begin{array}{ccc} |\epsilon_{kj}\rangle & =a_{1,2k+j+1}^{\left(w\right)}|00\rangle +{\left(-1\right)}^{k+j}a_{1,2k+j+1+{\left(-1\right)}^{j}}^{\left(w\right)}|01\rangle & \\ \; & \;\;+{\left(-1\right)}^{k}a_{1,2k+j+1+2{\left(-1\right)}^{k}}^{\left(w\right)}|10\rangle +{\left(-1\right)}^{j}a_{1,4-2k-j}^{\left(w\right)}|11\rangle , \end{array}$$
where k,j∈{0,1}. The coefficients $a_{1,n}^{\left(w\right)}$ represent participant-specific weights that contribute to the basis states. For the second particle pair $\left(N_{w}^{2},{\hat{N}}_{w}^{2}\right)$, the basis follows a similar structure to $|\epsilon_{kj}\rangle$, but with adjusted indices and coefficients. Specifically, the indices k,j are replaced with h,l, and the coefficients $a_{1,n}^{\left(w\right)}$ are replaced by $a_{2,n}^{\left(w\right)}$. For the third particle pair $\left(N_{w}^{3},{\hat{N}}_{w}^{3}\right)$, the measurement basis is given as(19)$$\begin{array}{ccc} |\omega_{st}^{\left(kj\right)}\rangle & =\frac{1}{2}[e^{-i\theta_{2k+j+1}^{w}}|00\rangle +{\left(-1\right)}^{s+t}e^{-i\theta_{2k+j+1+{\left(-1\right)}^{j}}^{w}}|01\rangle & \\ \; & \;\;+{\left(-1\right)}^{s}e^{-i\theta_{2k+j+1+2{\left(-1\right)}^{k}}^{w}}|10\rangle +{\left(-1\right)}^{t}e^{-i\theta_{4-2k-j}^{w}}|11\rangle ], \end{array}$$
where s,t∈{0,1}, and $e^{-i\theta_{n}^{w}}$ introduces specific phase adjustments for each basis state. For the fourth particle pair $\left(N_{w}^{4},{\hat{N}}_{w}^{4}\right)$, the measurement basis shares the same structure as $|\omega_{st}^{\left(kj\right)}\rangle$, but with indices k,j,s,t replaced by h,l,m,r, respectively. Similarly, the phase parameters $\theta_{n}^{w}$ are replaced with ${\hat{\theta }}_{n}^{w}$. Third, the supervisor Tom measures his particle *T* using the measurement operation ${|d\rangle }_{T}\langle d|$ (d∈{0,1}) and announces the measurement result ${|d\rangle }_{T}$ to all correspondents. Based on these results, all participants can reconstruct the target two-particle states. For simplicity, the intermediate measurement results, collapsed states, and corresponding recovery unitary operations are omitted here. Since the proposed extended scheme is similar in steps and operations to the four-party controlled bidirectional cyclic scheme for RSP presented in the previous section, the success probability of this extended scheme is 1.

## 4. Discussion and Conclusions

To the best of our knowledge, DDC quantum communication has been explored only in a few studies [41,42,44,45], which mainly address the mixed communication of single-particle states, RSP of single-particle states with real coefficients, and the remote preparation of dual-particle states with real coefficients. In conventional controlled schemes, a fixed set of measurement outcomes from the sender and supervisor is used to determine the necessary recovery transformations for the receiver. These approaches typically present the relationship between the measurement results and the corresponding recovery operations in a way that can be cumbersome, especially when extending to multiparty communication scenarios. The reliance on this rigid structure can hinder the scalability and generalizability of the schemes, making it less efficient for applications involving more complex quantum networks or multiparty communication. Unlike the schemes in the four references mentioned above, our scheme provides general analytical formulas applicable to dual-particle states, describing the unitary transformations performed by the sender, supervisor, and receiver. This overcomes the limitations of existing schemes, such as their weak reasoning ability and complex expressions, which are not ideal for future multi-particle quantum communication. Additionally, our scheme differs from the ones in the four references in the following ways: the scheme in Section 2 is novel and has not been previously reported, and in Section 3, we explore DDC controlled RSP for dual-particle states with complex coefficients, offering a more general and broader application potential than the scheme in reference [45]. This extension not only increases the applicability of the scheme but also enables it to handle more complex quantum communication network applications.

Intrinsic efficiency (IE) is an important metric for assessing the effectiveness of quantum communication protocols. It is defined as [13,46](20)$$\begin{array}{c} \omega =\frac{q_{t}}{b_{c}+q_{c}}, \end{array}$$
where $q_{t}$ is the number of qubits being transmitted, $q_{c}$ denotes the number of qubits in the quantum channel, and $b_{c}$ refers to the classical bits transferred. Through systematic optimization of channel utilization, our protocols achieve superior efficiency compared to existing schemes, as detailed in Table 1.

As demonstrated in Table 1, the two-particle scheme in Section 2.2 of our paper achieves a 40% improvement in IE compared to the single-particle scheme. The generalized scheme in Section 2.3 of our paper exhibits particularly noteworthy characteristics, approaching an asymptotic efficiency of $\omega_{\infty }=4/13$ as n→∞. This represents a 38% improvement over the corresponding single-particle state schemes, while maintaining linear scaling of resource requirements. The efficiency gains originate from optimized entanglement distribution strategies and reduced classical communication overhead through deterministic operator relationships. Further extension to complex Hilbert spaces yields additional protocol variants for complex-coefficient states. Specifically, the IEs of our protocols in Section 3.1 and Section 3.2 are $\omega =\frac{3}{16}$ and $\omega =\frac{4n}{21n+1}$, respectively. These results not only confirm that our framework improves upon existing real-coefficient implementations but also enable new capabilities for handling more sophisticated quantum states, further expanding the applicability of our approach. handling more sophisticated quantum states, further expanding the applicability of our approach.

Next, We briefly address the security of our protocols, which relies entirely on the secure pre-sharing of entanglement among the authorized participants. This refers to the security of the entangled resource during the distribution process. By using well-established and comprehensive inspection strategies [47,48] for other similar quantum tasks, any external malicious attack or internal deception is easily detectable. For simplicity, we omit further discussion on this. Therefore, we can conclude that our protocols are fully secure. Additionally, since all our schemes are controlled, it ensures that no communicator can reconstruct the desired states without the supervisor’s consent, thereby providing an extra layer of security. The security analysis is consistent with previous findings on entanglement robustness in noisy environments. As demonstrated by Hu [49], the decoherence characteristics of multipartite entangled states directly determine their viability as quantum channels. Our protocol relies on pre-shared entanglement resources, which aligns with their conclusions about entanglement persistence under controlled conditions. Furthermore, as shown by Jung et al. [50], the choice of entanglement structure plays a key role in the robustness of quantum teleportation through noisy channels. This further reinforces the reliability of the entanglement resources we depend on in practical quantum communication scenarios.

Taking the scheme outlined in Section 3.1 as an example, we examine the control power [45,51,52] of the supervisor David [38,42]. Suppose Alice’s measurement outcomes are $|\epsilon_{00}\rangle_{A_{1}^{\prime }A_{2}^{\prime }}{|\omega_{01}^{\left(00\right)}\rangle }_{A_{1}^{\prime }A_{2}^{\prime }}$ and $|\zeta_{01}\rangle_{A_{3}^{\prime }A_{4}^{\prime }}{|\varsigma_{00}^{\left(01\right)}\rangle }_{A_{3}^{\prime }A_{4}^{\prime }}$, Bob’s results are $|\epsilon_{00}^{\prime }\rangle_{B_{1}^{\prime }B_{2}^{\prime }}{|{\hat{\omega }}_{10}^{\left(00\right)}\rangle }_{B_{1}^{\prime }B_{2}^{\prime }}$ and $|\zeta_{00}^{\prime }\rangle_{B_{3}^{\prime }B_{4}^{\prime }}{|{\hat{\varsigma }}_{00}^{\left(00\right)}\rangle }_{B_{3}^{\prime }B_{4}^{\prime }}$, and Charlie’s outcomes are $|\epsilon_{10}^{\prime \prime }\rangle_{C_{1}^{\prime }C_{2}^{\prime }}{|{\bar{\omega }}_{00}^{\left(10\right)}\rangle }_{C_{1}^{\prime }C_{2}^{\prime }}$ and $|\zeta_{00}^{\prime \prime }\rangle_{C_{7}^{\prime }C_{8}^{\prime }}{|{\bar{\varsigma }}_{01}^{00}\rangle }_{C_{7}^{\prime }C_{8}^{\prime }}$. Based on Equation (A5) from Appendix A, it can be deduced that the entire system state will collapse into(21)$$\begin{array}{ccc} \; & \frac{1}{256\sqrt{2}}\{(a_{11}e^{i\theta_{1}}|00\rangle -a_{12}e^{i\theta_{2}}|01\rangle +a_{13}e^{i\theta_{3}}|10\rangle -a_{14}e^{i\theta_{4}}{\left|11\rangle \right)}_{B_{5}^{\prime }B_{6}^{\prime }} & \\ \; & \;\;⊗(a_{22}e^{i\theta_{2}}|00\rangle +a_{21}e^{i\theta_{1}}|01\rangle -a_{24}e^{i\theta_{4}}|10\rangle -a_{23}e^{i\theta_{3}}{\left|11\rangle \right)}_{C_{7}^{\prime }C_{8}^{\prime }} & \\ \; & \;\;⊗(b_{11}e^{i\alpha_{1}}|00\rangle -b_{12}e^{i\alpha_{2}}|01\rangle -b_{13}e^{i\alpha_{3}}|10\rangle +b_{14}e^{i\alpha_{4}}{\left|11\rangle \right)}_{C_{5}^{\prime }C_{6}^{\prime }} & \\ \; & \;\;⊗(b_{21}e^{i\alpha_{1}}|00\rangle -b_{22}e^{i\alpha_{2}}|01\rangle +b_{23}e^{i\alpha_{3}}|10\rangle -b_{24}e^{i\alpha_{4}}{\left|11\rangle \right)}_{A_{7}^{\prime }A_{8}^{\prime }} & \\ \; & \;\;⊗(c_{13}e^{i\beta_{3}}|00\rangle -c_{14}e^{i\beta_{4}}|01\rangle -c_{11}e^{i\beta_{1}}|10\rangle +c_{12}e^{i\beta_{2}}{\left|11\rangle \right)}_{A_{5}^{\prime }A_{6}^{\prime }} & \\ \; & \;\;⊗(c_{21}e^{i\beta_{1}}|00\rangle +c_{22}e^{i\beta_{2}}|01\rangle +c_{23}e^{i\beta_{3}}|10\rangle +c_{24}e^{i\beta_{4}}{\left|11\rangle \right)}_{B_{7}^{\prime }B_{8}^{\prime }}{|0\rangle }_{D} & \\ \; & \;\;+(a_{11}e^{i\theta_{1}}|11\rangle +a_{12}e^{i\theta_{2}}|10\rangle -a_{13}e^{i\theta_{3}}|01\rangle -a_{14}e^{i\theta_{4}}{\left|00\rangle \right)}_{B_{5}^{\prime }B_{6}^{\prime }} & \\ \; & \;\;⊗(a_{22}e^{i\theta_{2}}|11\rangle -a_{21}e^{i\theta_{1}}|10\rangle +a_{24}e^{i\theta_{4}}|01\rangle -a_{23}e^{i\theta_{3}}{\left|00\rangle \right)}_{C_{7}^{\prime }C_{8}^{\prime }} & \\ \; & \;\;⊗(b_{11}e^{i\alpha_{1}}|11\rangle +b_{12}e^{i\alpha_{2}}|10\rangle +b_{13}e^{i\alpha_{3}}|01\rangle +b_{14}e^{i\alpha_{4}}{\left|00\rangle \right)}_{C_{5}^{\prime }C_{6}^{\prime }} & \\ \; & \;\;⊗(b_{21}e^{i\alpha_{1}}|11\rangle +b_{22}e^{i\alpha_{2}}|10\rangle -b_{23}e^{i\alpha_{3}}|01\rangle -b_{24}e^{i\alpha_{4}}{\left|00\rangle \right)}_{A_{7}^{\prime }A_{8}^{\prime }} & \\ \; & \;\;⊗(c_{13}e^{i\beta_{3}}|11\rangle +c_{14}e^{i\beta_{4}}|10\rangle +c_{11}e^{i\beta_{1}}|01\rangle +c_{12}e^{i\beta_{2}}{\left|00\rangle \right)}_{A_{5}^{\prime }A_{6}^{\prime }} & \\ \; & \;\;⊗(c_{21}e^{i\beta_{1}}|11\rangle -c_{22}e^{i\beta_{2}}|10\rangle -c_{23}e^{i\beta_{3}}|01\rangle +c_{24}e^{i\beta_{4}}{\left|00\rangle \right)}_{B_{7}^{\prime }B_{8}^{\prime }}{|1\rangle }_{D}. & \end{array}$$
After that, Alice, Bob, and Charlie implement suitable unitary operations:(22)$$\begin{array}{ccc} U_{tot} & =\left[\sigma_{B_{5}^{\prime }}^{\left(0,0\right)}⊗\sigma_{B_{6}^{\prime }}^{\left(1,1\right)}\right]⊗\left[\sigma_{C_{7}^{\prime }}^{\left(1,1\right)}⊗\sigma_{C_{8}^{\prime }}^{\left(0,1\right)}\right]⊗\left[\sigma_{C_{5}^{\prime }}^{\left(1,1\right)}⊗\sigma_{C_{6}^{\prime }}^{\left(1,1\right)}\right] & \\ \; & \;\;⊗\left[\sigma_{A_{7}^{\prime }}^{\left(0,0\right)}⊗\sigma_{A_{8}^{\prime }}^{\left(1,1\right)}\right]⊗\left[\sigma_{A_{5}^{\prime }}^{\left(1,0\right)}⊗\sigma_{A_{6}^{\prime }}^{\left(1,1\right)}\right]⊗\left[\sigma_{B_{7}^{\prime }}^{\left(0,0\right)}⊗\sigma_{B_{8}^{\prime }}^{\left(0,0\right)}\right]⊗\sigma_{D}^{\left(0,0\right)}, & \end{array}$$
and then the combined state in Equation (Equation 22) transforms into(23)$$\begin{array}{ccc} \; & {|F\rangle }_{B_{5}^{\prime }B_{6}^{\prime }C_{7}^{\prime }C_{8}^{\prime }C_{5}^{\prime }C_{6}^{\prime }A_{7}^{\prime }A_{8}^{\prime }A_{5}^{\prime }A_{6}^{\prime }B_{7}^{\prime }B_{8}^{\prime }D} & \\ \; & =\frac{1}{256\sqrt{2}}{(|\phi_{1}\rangle }_{B_{5}^{\prime }B_{6}^{\prime }}|\phi_{2}\rangle_{C_{7}^{\prime }C_{8}^{\prime }}|\psi_{1}\rangle_{C_{5}^{\prime }C_{6}^{\prime }}|\psi_{2}\rangle_{A_{7}^{\prime }A_{8}^{\prime }}|\chi_{1}\rangle_{A_{5}^{\prime }A_{6}^{\prime }}|\chi_{2}\rangle_{B_{7}^{\prime }B_{8}^{\prime }}{|0\rangle }_{D} & \\ \; & \;\;-|\phi_{1}^{\prime }\rangle_{B_{5}^{\prime }B_{6}^{\prime }}|\phi_{2}^{\prime }\rangle_{C_{7}^{\prime }C_{8}^{\prime }}|\psi_{1}^{\prime }\rangle_{C_{5}^{\prime }C_{6}^{\prime }}|\psi_{2}^{\prime }\rangle_{A_{7}^{\prime }A_{8}^{\prime }}|\chi_{1}^{\prime }\rangle_{A_{5}^{\prime }A_{6}^{\prime }}|\chi_{2}^{\prime }\rangle_{B_{7}^{\prime }B_{8}^{\prime }}{|1\rangle }_{D} & \end{array}$$
where $|\phi_{k}\rangle$, $|\psi_{k}\rangle$, and $|\chi_{k}\rangle$ (k=0,1) are the same as shown in Equation (Equation 10), and Alice’s state $|\phi_{1}^{\prime }\rangle$ can be expressed as$$\begin{array}{ccc} \; & |\phi_{1}^{\prime }\rangle =a_{11}e^{i\theta_{1}}|11\rangle -a_{12}e^{i\theta_{2}}|10\rangle -a_{13}e^{i\theta_{3}}|01\rangle +a_{14}e^{i\theta_{4}}|00\rangle . & \end{array}$$
The second state, $|\phi_{2}^{\prime }\rangle$, has a similar structure to $|\phi_{1}^{\prime }\rangle$ but is derived by replacing the coefficients $a_{1j}$ with $a_{2j}$ for j=1,2,3,4, while keeping the phase factors $\theta_{j}$ unchanged and preserving the same sign pattern. For Bob and Charlie, their states share a similar structure with $|\phi_{k}^{\prime }\rangle$ and can be derived by substituting coefficients and phase factors. Specifically, Bob’s states $|\psi_{k}^{\prime }\rangle$ can be obtained by replacing Alice’s coefficients $a_{ij}$ with $b_{ij}$ and the phase factors $\theta_{j}$ with $\alpha_{j}$, resulting in $|\psi_{1}^{\prime }\rangle$ and $|\psi_{2}^{\prime }\rangle$. Similarly, Charlie’s states $|\chi_{k}^{\prime }\rangle$ are derived by substituting $a_{ij}$ with $c_{ij}$ and $\theta_{j}$ with $\beta_{j}$, yielding $|\chi_{1}^{\prime }\rangle$ and $|\chi_{2}^{\prime }\rangle$. To avoid lengthy formulaic descriptions, the explicit expressions for Bob’s and Charlie’s states are omitted here.

If the supervisor David does not grant permission, the state of the system after measurements by Alice, Bob, and Charlie collapses into a mixed state. This mixed state can be represented using a density operator:$$\begin{array}{cc} \rho_{B^{\prime }C^{\prime }A^{\prime }} & =tr_{D}\left(\rho_{B^{\prime }C^{\prime }A^{\prime }D}\right)=tr_{D}(|F\rangle \langle F|)=\frac{1}{131072}\left[|\Phi_{1}\rangle \langle \Phi_{1}\left|+\right|\Phi_{2}\rangle \langle \Phi_{2}|\right], \end{array}$$
where$$\begin{array}{c} |\Phi_{1}\rangle =|\phi_{1}\rangle_{B_{5}^{\prime }B_{6}^{\prime }}⊗|\phi_{2}\rangle_{C_{7}^{\prime }C_{8}^{\prime }}⊗|\psi_{1}\rangle_{C_{5}^{\prime }C_{6}^{\prime }}⊗|\psi_{2}\rangle_{A_{7}^{\prime }A_{8}^{\prime }}⊗|\chi_{1}\rangle_{A_{5}^{\prime }A_{6}^{\prime }}⊗{|\chi_{2}\rangle }_{B_{7}^{\prime }B_{8}^{\prime }}. \end{array}$$
The state $|\Phi_{2}\rangle$ shares a similar structure to $|\Phi_{1}\rangle$. Specifically, $|\Phi_{2}\rangle$ is obtained by replacing the states $|\phi_{k}\rangle$, $|\psi_{k}\rangle$, and $|\chi_{k}\rangle$ in $|\Phi_{1}\rangle$ with their primed counterparts $|\phi_{k}^{\prime }\rangle$, $|\psi_{k}^{\prime }\rangle$, and $|\chi_{k}^{\prime }\rangle$, respectively. The density matrix $\rho_{B^{\prime }C^{\prime }A^{\prime }}=\rho_{B_{5}^{\prime }B_{6}^{\prime }C_{7}^{\prime }C_{8}^{\prime }C_{5}^{\prime }C_{6}^{\prime }A_{7}^{\prime }A_{8}^{\prime }A_{5}^{\prime }A_{6}^{\prime }B_{7}^{\prime }B_{8}^{\prime }}$ is composed of two orthogonal states, $|\Phi_{1}\rangle$ and $|\Phi_{2}\rangle$, combined through their outer products. Here, $tr_{D}\left(\cdot \right)$ denotes the partial trace over particle *D*. In this way, we are able to calculate the average fidelity of the composite state shared among the three communicators and deduce David’s control capability as follows (for the detailed calculation process, please refer to Appendix B):$${\bar{f}}_{ABC}=\frac{1}{131072}\left[1+{\left(\frac{1}{6}\right)}^{6}\right]$$
and$$P_{D}=1-{\bar{f}}_{ABC}=1-\frac{1}{131072}\left[1+{\left(\frac{1}{6}\right)}^{6}\right],$$
respectively.

As highlighted in reference [52], the supervisor’s control power must satisfy the following condition:$$P\geq \frac{2^{N}-1}{2^{N}+1},$$
where *N* represents the number of qubits being teleported. It can be readily verified that the control power achieved by David in Section 3.1 fulfils $P_{D}>\left(2^{12}-1\right)/\left(2^{12}+1\right)$. This demonstrates that, from the supervisor’sperspective, the protocol described in Section 3.1 is both reasonable and feasible.

Turning our attention to the feasibility of the proposed schemes, it is evident that their implementation involves fundamental quantum operations, including Bell-state measurements, single-particle measurements, and the application of quantum gates such as the Hadamard gate, CNOT gate, and Pauli gates. These operations have been successfully realized across a variety of experimental quantum platforms, including the cavity QED system [53], ion trap system [54], and optical systems [55], among others. Given the maturity of these technologies, the protocols proposed in this work are experimentally viable and can be implemented with current advancements in quantum technology.

In summary, in this work, we constructed a 25-particle entangled state based on Hadamard and CNOT gates to serve as a quantum channel, and on this basis, proposed two novel four-party CDDC schemes tailored for QT and RSP. In the QT scheme, under the controller’s authorization, three communicators each transmit two arbitrary unknown two-particle states to the other two communicators. In this process, each communicator performs only four Bell-state measurements, while, with the controller’s approval, a single-particle Z-basis measurement is performed to achieve bidirectional transmission of the two unknown two-particle states. The receivers can deterministically reconstruct the target states by selecting the corresponding unitary operations based on the measurement outcomes. For the RSP scenario, under the supervision of the controller, each communicator, with the assistance of the other communicators, can prepare two different arbitrary two-particle states. To accomplish this, each communicator introduces four auxiliary qubits and performs four CNOT gate operations, and then, by combining a feedforward strategy, cleverly constructs different measurement bases to perform two two-particle measurements. Subsequently, the controller measures their particle in the Z-basis, after which each receiver perfectly recovers the target state by applying the appropriate Pauli operations based on the measurement results of the other three participants. Both schemes achieve a theoretical success rate of 100%. Furthermore, we have extended the two proposed four-party CDDC quantum communication schemes to the case of *n* communicators and one supervisor, i.e., an n+1 party scenario, where *n* is greater than 3. Since the extended scheme follows a similar procedure and operational approach as the previous schemes—wherein each communicator, under supervisory control, can simultaneously transmit two different arbitrary two-particle states to the other parties, thereby achieving controlled quantum cyclic communication in both clockwise and counterclockwise directions and ultimately realizing a 100% success rate—we provide a general mathematical formulation for bidirectional cyclic quantum communication with multi-particle states applicable to each scheme. This offers a scalable operational framework for multi-particle bidirectional cyclic communication. Moreover, we evaluated the inherent efficiency, security, and controllability of the proposed schemes. Compared with previous studies, our schemes are efficient, controllable, secure, and experimentally feasible.
With the continuous advancement of quantum technologies, the schemes proposed in this study can be further expanded to meet more complex communication requirements. Through innovations in communication protocols, mathematical formulations, and the extension of the schemes, this research provides theoretical and technical support for the ongoing development of multi-particle quantum communication, aiming to enhance security, capacity, and meet the diverse needs of future quantum network scenarios.

## Figures and Tables

**Figure 1 entropy-27-00292-f001:**
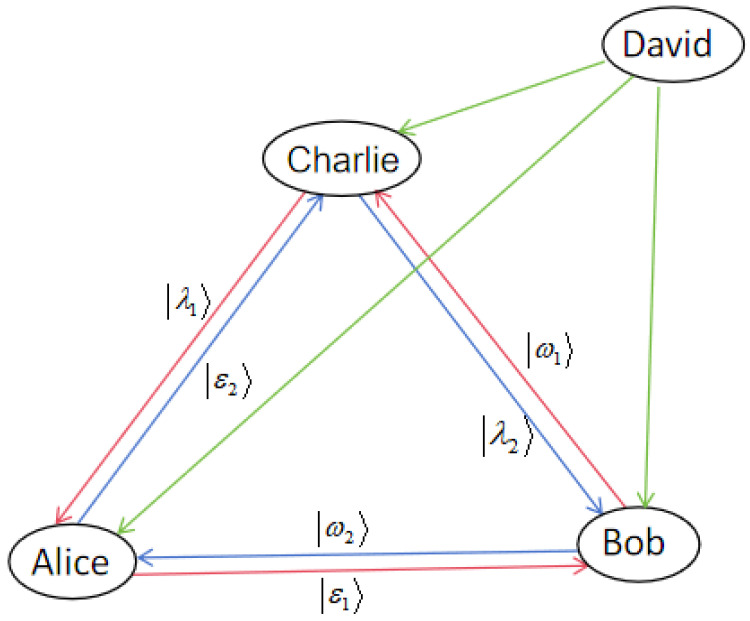
The relationship between the three communicators and one supervisor. The blue and red lines with arrows represent the quantum states to be transmitted, while the green straight line with an arrow represents the transmission of supervisor information.

**Figure 2 entropy-27-00292-f002:**
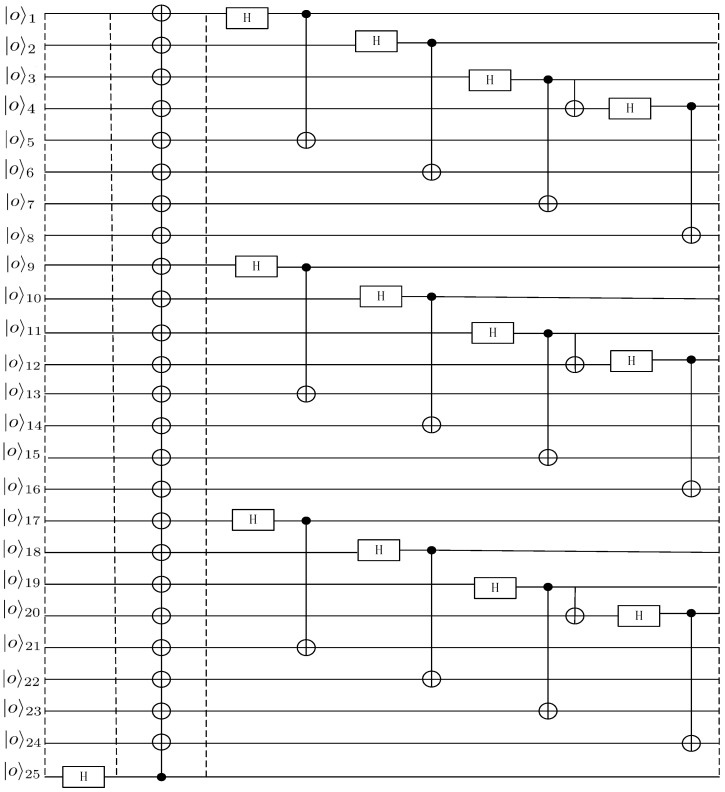
The quantum circuit for constructing the 25-qubit entangled channel, where *H* represents the Hadamard gate operation, and the solid black “·” and “⊕” together form a CNOT operation. The Hadamard gate is used to transform the qubit into a superposition state, while the CNOT gate is used to create entanglement between two qubits.

**Figure 3 entropy-27-00292-f003:**
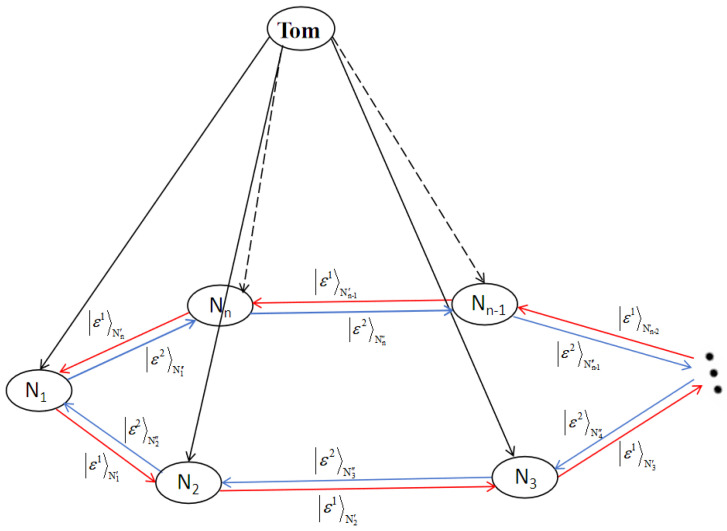
The relationship between *n* communicators and one supervisor. The red and blue straight lines with arrows represent the quantum states to be transmitted, while the black straight line with an arrow represents the transmission of supervisor information.

**Table 1 entropy-27-00292-t001:** Comparison of IE for DDC quantum communication schemes.

Scheme	$q_{t}$	$q_{c}+b_{c}$	ω	$\omega_{\infty }$
Ref. [44] (Single-particle)	6	13 + 15	3/14≈0.214	–
Ref. [44] (Multiparty)	2n	(4n+1)+5n	2n/(9n+1)	2/9≈0.222
the schemes in Section 2.2	12	25 + 15	3/10=0.3	–
the schemes in Section 2.3	4n	(8n+1)+5n	4n/(13n+1)	4/13≈0.308

## Data Availability

The data sets generated during and/or analysed during the current study are available from the corresponding author on reasonable request.

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
