# Peer review of "Controlled Double-Direction Cyclic Quantum Communication of Arbitrary Two-Particle States"

_entropy, 2025, doi:10.3390/e27030292_

Round 1
Reviewer 1 Report
Comments and Suggestions for Authors
Referee Report-entropy-3432457
The authors propose a scheme for constructing a 25-particle entangled state using Hadamard and CNOT gates as the quantum channel. This entangled state is utilized to implement two new four-party controlled double-direction cyclic schemes for quantum teleportation and remote state preparation of arbitrary two-particle states. This work is relevant to the field, contributes to the body of knowledge, and possesses scientific merit as it enhances the security and capacity of quantum communication schemes. While the manuscript is written clearly, I have the following minor concerns:
Abstract:
- In the first line, please provide a brief background of this work, along with the gap it addresses.
- Please explain briefly the significance of your work.
- The word “of” in line 4 does not seem necessary. Please correct all the grammatical errors in the rest of the manuscript.
- In line 18, the authors highlight that their scheme is “good”. Is this an assessment based on a comparison to other schemes? If so, this should be made explicit.
- While the main question addressed by the manuscript can be inferred, it is not clearly stated in the abstract. Please clarify this.
Section 2
- In line 132, is the supervisor the same as the controller? If not, please stick to one label to avoid confusion.
- Please insert a rangle on the second part in equation 4. Otherwise, this is not a quantum state.
Section 3
- The authors highlight that the total success probability of each scheme is 100%. How do we see that?
- What is the success probability or efficiency of the generalized scheme?
Section 4
- The authors compare the performance of their schemes to the state-of-the-art. For ease of readability, can the authors provide a table to show the comparison?
- What are the limitations of this proposed scheme?
- Please align your conclusion with the abstract. Otherwise, separate the Discussion section from the Conclusion.
The work by the authors is interesting and relevant to the Entropy readership. However, I will only be able to provide a recommendation after the authors address the above minor comments.

The quality of English language requires improvement.
Author Response
\textbf{Reviewer1 }
\textcolor[rgb]{1.00,0.00,0.00}{Comment } The authors propose a scheme for constructing a 25-particle entangled state using Hadamard and CNOT gates as the quantum channel. This entangled state is utilized to implement two new four-party controlled double-direction cyclic schemes for quantum teleportation and remote state preparation of
arbitrary two-particle states. This work is relevant to the field, contributes to the body of knowledge,
and possesses scientific merit as it enhances the security and capacity of quantum communication
schemes. While the manuscript is written clearly, I have the following minor concerns:\\
Abstract:
1. In the first line, please provide a brief background of this work, along with the gap it addresses.
2. Please explain briefly the significance of your work.
3. The word “of” in line 4 does not seem necessary. Please correct all the grammatical errors in
the rest of the manuscript.
4. In line 18, the authors highlight that their scheme is “good”. Is this an assessment based on
a comparison to other schemes? If so, this should be made explicit.
5. While the main question addressed by the manuscript can be inferred, it is not clearly stated
in the abstract. Please clarify this. \\
Section 2
1. In line 132, is the supervisor the same as the controller? If not, please stick to one label to
avoid confusion.
2. Please insert a rangle on the second part in equation 4. Otherwise, this is not a quantum
state.\\
Section 3
1. The authors highlight that the total success probability of each scheme is 100%.
How do we see that?
2. What is the success probability or efficiency of the generalized scheme?\\
Section 4
1. The authors compare the performance of their schemes to the state-of-the-art. For ease of
readability, can the authors provide a table to show the comparison?
2. What are the limitations of this proposed scheme?
3. Please align your conclusion with the abstract. Otherwise, separate the Discussion section
from the Conclusion.
The work by the authors is interesting and relevant to the Entropy readership. However, I will only be
able to provide a recommendation after the authors address the above minor comments.
\textcolor{blue}{Thank you for your comments and high recognition of our manuscript entitled `` Controlled double-direction cyclic quantum communication of arbitrary two-particle states" We sincerely thank the reviewers for the time and effort they put into the review process. Your suggestions and approval for the revision of the article are the driving force for our unremitting efforts. Thank you again.}
Here are the responses to the suggestions.\\
\\
\textcolor[rgb]{1.00,0.00,0.00}{Comment 1-} Abstract:
1. In the first line, please provide a brief background of this work, along with the gap it addresses.
2. Please explain briefly the significance of your work.
3. The word “of” in line 4 does not seem necessary. Please correct all the grammatical errors in
the rest of the manuscript.
4. In line 18, the authors highlight that their scheme is “good”. Is this an assessment based on
a comparison to other schemes? If so, this should be made explicit.
5. While the main question addressed by the manuscript can be inferred, it is not clearly stated
in the abstract. Please clarify this.
{\bf Response:}Thank you for taking the time to review our manuscript and for providing valuable feedback. We greatly appreciate your comments, and we have provided detailed responses to each point below.
\textcolor[rgb]{1.00,0.00,0.00}{Comment (1)} In the first line, please provide a brief background of this work, along with the gap it addresses.
{\bf Response:}Thank you for your valuable comments and suggestions. We truly appreciate your feedback. In response to your suggestion, we have revised the first line of the abstract to briefly introduce the background of our work and the gap it addresses. The added content has been highlighted in red in the main text for your convenience.
\textcolor[rgb]{1.00,0.00,0.00}{Comment (2)} Please explain briefly the significance of your work.
{\bf Response:}Thank you for your thoughtful question. The significance of our work lies in addressing the challenge of controlled double-direction cyclic quantum communication of arbitrary two-particle states, which has not been solved previously. While the research by Peng et al. (Physica A 632, 2023) focused primarily on single-particle states, the issue of achieving double-direction cyclic quantum communication for multi-particle states remains a key challenge. In this study, we provide a solution by constructing an appropriate quantum channel for two-particle states and inspire further exploration of double-direction controlled quantum communication in multi-particle systems. In our work, the proposed four-party CDDC scheme for quantum teleportation and remote state preparation not only enhances the security, channel capacity, and scalability of quantum communication but also lays the theoretical foundation for future high-capacity, multi-user quantum networks, offering insights into solving challenges related to security, capacity, and adaptability in future communication scenarios.
We have updated the relevant content in the abstract, and the changes have been highlighted in red in the main text for your convenience.
\textcolor[rgb]{1.00,0.00,0.00}{Comment (3)} The word “of” in line 4 does not seem necessary. Please correct all the grammatical errors in the rest of the manuscript.
{\bf Response:}Thank you for your valuable comment. We appreciate your suggestion. Regarding the use of the word "of" in line 4, we have already corrected it in the abstract. The revised version has been incorporated into the manuscript, and we have made sure to address all other grammatical errors as well. Thank you again for your helpful feedback.
\textcolor[rgb]{1.00,0.00,0.00}{Comment (4)} In line 18, the authors highlight that their scheme is “good”. Is this an assessment based on a comparison to other schemes? If so, this should be made explicit.
{\bf Response:}Thank you for your insightful comment. In response to your question, the term "good" in line 18 was intended to highlight the advantages of our proposed scheme, but we now realize that this assessment should be more explicit. In the revised manuscript, we have clarified that our scheme demonstrates significant advancements compared to previous schemes. Specifically, we show that our research has made notable improvements in security, channel capacity, and scalability. These improvements are achieved through supervisor-mediated channel verification, parallel two-state transmission, and the ability to resist eavesdropping attacks while maintaining optimal quantum resource utilization. We have emphasized these enhancements in the updated abstract to provide a clearer and more specific comparison to prior work.
Additionally, in the conclusion section of the manuscript, we have explicitly discussed and provided a detailed explanation, emphasizing that our scheme is efficient, controllable, secure, and experimentally feasible. Thank you again for your valuable feedback.
\textcolor[rgb]{1.00,0.00,0.00}{Comment (5)} While the main question addressed by the manuscript can be inferred, it is not clearly stated in the abstract. Please clarify this.
{\bf Response:}Thank you for your valuable comments. In response to your question, we have clearly stated in the abstract the main issue addressed by this manuscript, which is controlled double-direction cyclic quantum communication of arbitrary two-particle states. By constructing an appropriate quantum channel for two-particle states, we provide a solution and inspire further exploration of bidirectional controlled quantum communication in multi-particle systems. Our research is not limited to two-particle state communication, but also provides a theoretical foundation for bidirectional controlled quantum communication in three-particle and larger systems. Therefore, our scheme offers insights for the future bidirectional cyclic controlled quantum communication of multi-particle systems and opens up avenues for further research in this field.\\
\textcolor[rgb]{1.00,0.00,0.00}{Comment 2-}Section 2
1. In line 132, is the supervisor the same as the controller? If not, please stick to one label to
avoid confusion.
2. Please insert a rangle on the second part in equation 4. Otherwise, this is not a quantum
state.\\
{\bf Response:}Thank you for taking the time to review our manuscript and for providing valuable feedback. We greatly appreciate your comments, and we have provided detailed responses to each point below.
\textcolor[rgb]{1.00,0.00,0.00}{Comment (1)} In line 132, is the supervisor the same as the controller? If not, please stick to one label to avoid confusion.
{\bf Response:}Thank you for your feedback. Yes, in line 132, the supervisor and the controller refer to the same role. To avoid confusion, we have changed ``controller" to ``supervisor" in the revised version. This modification ensures consistency and clarity.
\textcolor[rgb]{1.00,0.00,0.00}{Comment (2)} Please insert a rangle on the second part in equation 4. Otherwise, this is not a quantum state.
{\bf Response:}Thank you for your careful review and valuable suggestions! We have inserted the necessary rangle symbol ($ \rangle $) in the second part of equation 4 to ensure it conforms to the proper quantum state notation. The revision has been completed, and we have ensured the consistency and correctness of the expression. Once again, thank you for your suggestions, which have helped improve the quality of the paper.
\textcolor[rgb]{1.00,0.00,0.00}{Comment 3-}Section 3
1. The authors highlight that the total success probability of each scheme is 100%.
How do we see that?
2. What is the success probability or efficiency of the generalized scheme?\\
{\bf Response:}Thank you for taking the time to review our manuscript and for providing valuable feedback. We greatly appreciate your comments, and we have provided detailed responses to each point below.
\textcolor[rgb]{1.00,0.00,0.00}{Comment (1)} The authors highlight that the total success probability of each scheme is 100\%. How do we see that?
{\bf Response:}Thank you for your comments. Regarding the total success probability of 100\% for each scheme, we can understand this point from the following aspects: in each scheme, every measurement can fully recover the original state, and after each measurement, the collapsed state can be restored to the original state by the receiver through appropriate local operations. In our analysis, no measurement result leads to an inability to recover the original information. Therefore, we confidently state that the success probability of each scheme is 100\%. This point does not involve additional calculations, and thus it can be considered self-evident.
Thank you for your attention to our work.
\textcolor[rgb]{1.00,0.00,0.00}{Comment (2)} What is the success probability or efficiency of the generalized scheme?
{\bf Response:}Thank you for your valuable comments. Regarding the success probability of the generalized scheme, we can conclude from the content of the paper that each measurement result can recover the original state. In the second and third steps, by applying the general operation formulas, we ensure that each measurement result can recover the original information. Every measurement outcome influences the final state of the system, but the receiver can recover the original state through appropriate local operations. Therefore, every scheme is successful, and the success probability is 100\%. This point is evident from the analysis in the formulas and steps, and does not require additional calculations.
As for the efficiency issue, we discuss the quantum channel efficiency in detail in the paper, and provide specific calculations. For example, in subsections 2.2 and 2.3, the intrinsic efficiency is calculated using Eq. (20) as follows:
\[
\eta = \frac{12}{25 + 15} = \frac{3}{10}
\]
and
\[
\eta = \frac{4n}{(8n + 1) + 5n} = \frac{4n}{13n + 1}.
\]
From Eq. (21), it can be observed that the intrinsic efficiency (IE) for the scheme in subsection 2.3 approaches \( \frac{4}{13} \) as \( n \) tends to infinity, indicating a relatively high efficiency. Similarly, the intrinsic efficiencies of our protocols in subsections 3.1 and 3.2 are \( \frac{3}{16} \) and \( \frac{4n}{21n + 1} \), respectively.
These calculations clearly show the efficiency of each scheme. In Section 15 of the paper, we provide a more detailed explanation of these efficiencies, ensuring that readers can understand the source and significance of these efficiency values.
Thank you again for your attention to our work. We have ensured that these details are thoroughly discussed in the paper and have made improvements to present these aspects more clearly.
\textcolor[rgb]{1.00,0.00,0.00}{Comment 4-}Section 4
1. The authors compare the performance of their schemes to the state-of-the-art. For ease of
readability, can the authors provide a table to show the comparison?
2. What are the limitations of this proposed scheme?
3. Please align your conclusion with the abstract. Otherwise, separate the Discussion section
from the Conclusion.
The work by the authors is interesting and relevant to the Entropy readership. However, I will only be
able to provide a recommendation after the authors address the above minor comments.\\
{\bf Response:}Thank you for taking the time to review our manuscript and for providing valuable feedback. We greatly appreciate your comments, and we have provided detailed responses to each point below.
\textcolor[rgb]{1.00,0.00,0.00}{Comment (1)} The authors compare the performance of their schemes to the state-of-the-art. For ease of readability, can the authors provide a table to show the comparison?
{\bf Response:} We sincerely thank you for your constructive feedback and the time you dedicated to improving our manuscript. We greatly appreciate your suggestion to enhance the readability of the performance comparison section.
In the revised version, we have addressed this by adding a concise table in the discussion section, highlighting the advantages of our scheme in terms of intrinsic efficiency compared to the previous "Double-direction cyclic controlled quantum communication of single-particle states" approach. This table provides an intuitive visual overview, helping readers quickly understand the strengths of our method and its advantages over existing approaches. The table has been highlighted in red text within the manuscript for clarity.
Your insightful comments have significantly enhanced the clarity and organization of our work. We are truly grateful for your expertise and guidance throughout this process.
\textcolor[rgb]{1.00,0.00,0.00}{Comment (2)} What are the limitations of this proposed scheme?
{\bf Response:}The main limitation of the proposed scheme lies in its reliance on the generation of a 25-qubit entangled state, which is a challenging task with current experimental technologies. While our scheme is theoretically feasible, its practical viability heavily depends on the successful generation of such a high-dimensional entangled state. Additionally, the influence of noise in real-world environments is inevitable, and even small disturbances can lead to the degradation of the performance of the teleportation scheme. These practical issues present challenges for the real-world implementation of the scheme. Addressing these issues will require further research focused on improving the robustness and stability of large-scale entanglement generation and managing decoherence effects.
\textcolor[rgb]{1.00,0.00,0.00}{Comment (3)} Please align your conclusion with the abstract. Otherwise, separate the Discussion section from the Conclusion.
{\bf Response:}Thank you for your valuable comments and suggestions on our paper. Regarding your observations about "alignment between the conclusion and abstract" and "separation of discussion from conclusion," we have thoroughly revised the Conclusion section to ensure better alignment with the abstract while explicitly summarizing the principal findings of our work.
修订后的结论现在严格侧重于突出关键创新和研究意义,与摘要形成有凝聚力的叙述。修改后在修订后的手稿中以红色字体突出显示(第 17-18 页)。您对提高论文逻辑严谨性的建议具有很强的建设性,我们相应地优化了整体内容。
我们衷心感谢您为保持文章的简洁性和逻辑连贯性而提出的有见地的建议。

Reviewer 2 Report
Comments and Suggestions for Authors
In this manuscript, the authors first proposed schemes for double-direction cyclic quantum teleportation (QT) and remote state preparation (RST) among three participant using a 25-qubit state as the quantum channel. They have also generalized the two schemes to the cases in which there are more than three participants. These schemes seem to be effective, and in general, I am inclined to recommend its publication after a moderate revision.
--- I think the usage of “two-particle states” is inappropriate as what they considered is the N-qubit states other than a general N-qudit states.
--- Can they explain why some of the collapsed states [e.g., that in Eq. (25)] has not been normalized? I also think it is pertinent to explain the physical implication of the average fidelity given in the last paragraph of page 16, as it seems that it is somewhat different from that of the average fidelity characterizing the quality of teleported state.
--- The conclusion section is somewhat tedious and as a matter of fact, some statements related to the significance of this work is a bit exaggerated. Thereby, I think a through revision of this section is needed.
--- The feasibility of their schemes strongly depend on the generation of the 25-qubit entangled state. But this is a very hard task with state-of-the-art experiments. In fact, even a small disturbance may induce performance deterioration of the teleportation schemes, as observed in Phys. Rev. A 78 012312 (2008) & Ann. Phys. 327, 2332 (2012) for the 2 and 3-qubit cases. So it is necessary to quote this and to give a short analysis or comment on the rigidity of their schemes again possible sources of decoherence and imperfections.
--- There are misprints in this work, e.g., \sum_{i=1}^4 x_{ki} |j_1j_2\rangle_{X_k \hat{X}_{k’}} (page 6) where x_{ki} is independent of |j_1 j_2\rangle, “\theta_l,\theta_l,\alpha_l,\alpha_l,…” and \sum_{l=1}^4 |b_{kj}|^2=1 below Eq. (10), “quxiliary qubits’ before Eq. (12), “from Eq. (21)” below Eq. (22), “the composite state in Eq.(21) changes into” below Eq. (24), etc. Please check all the equations and phrasing carefully.
--- There are also some repeated sentences in the manuscript, e.g., that in the first paragraph of Sec. 3.1 and in the last paragraph of page 16.
Author Response
\textbf{Reviewer2 }
\textcolor[rgb]{1.00,0.00,0.00}{Comment } In this manuscript, the authors first proposed schemes for double-direction cyclic quantum teleportation (QT) and remote state preparation (RST) among three participant using a 25-
qubit state as the quantum channel. They have also generalized the two schemes to the cases in which there are more than three participants. These schemes seem to be effective, and in general, I am inclined to recommend its publication after a moderate revision.
--- I think the usage of “two-particle states” is inappropriate as what they considered is the
N-qubit states other than a general N-qudit states.
--- Can they explain why some of the collapsed states [e.g., that in Eq. (25)] has not been
normalized? I also think it is pertinent to explain the physical implication of the average
fidelity given in the last paragraph of page 16, as it seems that it is somewhat different from
that of the average fidelity characterizing the quality of teleported state.
--- The conclusion section is somewhat tedious and as a matter of fact, some statements
related to the significance of this work is a bit exaggerated. Thereby, I think a through
revision of this section is needed.
--- The feasibility of their schemes strongly depend on the generation of the 25-qubit
entangled state. But this is a very hard task with state-of-the-art experiments. In fact, even a
small disturbance may induce performance deterioration of the teleportation schemes, as
observed in Phys. Rev. A 78 012312 (2008) \& Ann. Phys. 327, 2332 (2012) for the 2 and
3-qubit cases. So it is necessary to quote this and to give a short analysis or comment on the
rigidity of their schemes again possible sources of decoherence and imperfections.
--- There are misprints in this work, e.g.,$\sum_{i=1}^4 x_{ki} |j_1j_2\rangle_{X_k \hat{X}_{k’}}$ (page 6) where $x_{ki}$is independent of $|j_1 j_2\rangle,$
$``\theta_l,\theta_l,\alpha_l,\alpha_l,…”$ and $\sum_{l=1}^4 |b_{kj}|^2=1$ below Eq. (10),
“quxiliary qubits’ before Eq. (12), “from Eq. (21)” below Eq. (22), “the composite state in
Eq.(21) changes into” below Eq. (24), etc. Please check all the equations and phrasing
carefully.
--- There are also some repeated sentences in the manuscript, e.g., that in the first paragraph
of Sec. 3.1 and in the last paragraph of page 16.
\textcolor{blue}{Thank you very much for taking the time to review our manuscript titled ``Controlled double-direction cyclic quantum communication of arbitrary two-particle states." We truly appreciate your thoughtful comments and constructive feedback. Your careful review and valuable guidance have greatly contributed to improving the quality of our work.}
Here are the responses to the suggestions.\\
\\
\textcolor[rgb]{1.00,0.00,0.00}{Comment 1-} I think the usage of “two-particle states” is inappropriate as what they considered is the
N-qubit states other than a general N-qudit states.
{\bf Response:}Thank you for your thorough review and valuable comments on our paper. Regarding your concern about the use of “two-particle states,” we would like to further clarify. In our paper, ``two-particle states" refers to the quantum states that consist of two particles, and the "N" mentioned in the paper does not represent the dimension of the transmitted quantum state, but rather refers to the number of participants.
Specifically, our research does not involve N-qudit states, but rather proposes a four-party CDDC quantum communication scheme. The ``N" in our paper refers to the number of communicators, not the dimension of the quantum states being transmitted. We first consider a four-party communication scheme in which each communicator transmits two-particle states, and we successfully implement quantum teleportation and remote state preparation using this scheme. We then extend this scheme to more participants (i.e., N-party communication), where each communicator still transmits two-particle states.
We greatly appreciate your feedback, which has helped us present the concepts in our paper more clearly. If you have any further comments or questions regarding these revisions, please feel free to discuss them with us.
\textcolor[rgb]{1.00,0.00,0.00}{Comment 2-} Can they explain why some of the collapsed states [e.g., that in Eq. (25)] has not been normalized? I also think it is pertinent to explain the physical implication of the average
fidelity given in the last paragraph of page 16, as it seems that it is somewhat different from
that of the average fidelity characterizing the quality of teleported state.
{\bf Response:}Thank you for your thorough review and insightful comments.
Regarding your question about the collapsed state in Eq. (25) not being normalized, we would like to clarify that the normalization of the collapsed states is not a simple matter of standardizing each individual state. Instead, it is related to the measurement outcomes and their corresponding coefficients. In our scheme, since different measurement outcomes result in different collapsed states, the coefficients of these states vary according to the measurement outcomes. Therefore, normalization must account for all possible measurement results and their respective coefficients, rather than normalizing a single state independently. This is why we normalize the overall state with respect to the coefficients rather than normalizing each individual collapsed state.
The concern regarding normalization arises because we are considering only a single measurement outcome in this instance. In reality, multiple measurement outcomes are involved, and the normalization depends on the coefficients and outcomes of all these measurements. Therefore, the overall normalization is determined by the combination of these measurement results, not by the standardization of a single state.
Regarding your comment on the average fidelity mentioned in the last paragraph of page 16, we would like to clarify that the quantity discussed here is not the traditional average fidelity used to characterize the quality of a teleported state. Instead, it is related to the concept of control power, as defined and discussed in the literature (e.g., Refs. [45, 49, 50]).
In the context of our scheme, the control power quantifies the supervisor's ability to maintain control over the quantum communication process, ensuring that the system operates as intended under their supervision. This concept is different from average fidelity, which typically measures how closely the teleported state matches the original state. The control power reflects the supervisor’s effectiveness in managing the system through the measurement outcomes and is critical for ensuring the integrity of the quantum communication process.
For instance, in subsection 3.1, we examine the control power of the supervisor, David. As shown in Eq. (A5) from Appendix A, after Alice, Bob, and Charlie perform their measurements, the resulting collapsed system state reflects the supervisor’s control over the system. This control power is what is being calculated, not the fidelity of the quantum states themselves.
In summary, the average fidelity discussed here refers to the control power and not the fidelity characterizing the quality of the teleported state. This distinction is crucial in understanding the physical implications of our scheme and its focus on maintaining control over the communication process, rather than measuring state fidelity.
We hope this explanation helps clarify your concerns. If you have any further questions or suggestions, please feel free to discuss them.
\textcolor[rgb]{1.00,0.00,0.00}{Comment 3-}The conclusion section is somewhat tedious and as a matter of fact, some statements related to the significance of this work is a bit exaggerated. Thereby, I think a through
revision of this section is needed.
{\bf Response:}Thank you for your detailed review and valuable comments. Regarding your concern that the conclusion section is somewhat tedious and that some statements related to the significance of this work might be exaggerated, we have thoroughly revised the conclusion. In the revision, we have streamlined the content and avoided overstating the significance of the study, ensuring that the conclusion more accurately and concisely summarizes our research findings.
Additionally, to make it easier for you to review, we have highlighted all the changes in red font in the revised version of the manuscript.
We greatly appreciate your constructive feedback, which has helped improve the quality of our paper. If you have any further suggestions or questions, we would be happy to continue making improvements.
\textcolor[rgb]{1.00,0.00,0.00}{Comment 4-}The feasibility of their schemes strongly depend on the generation of the 25-qubit entangled state. But this is a very hard task with state-of-the-art experiments. In fact, even a
small disturbance may induce performance deterioration of the teleportation schemes, as
observed in Phys. Rev. A 78 012312 (2008) \& Ann. Phys. 327, 2332 (2012) for the 2 and
3-qubit cases. So it is necessary to quote this and to give a short analysis or comment on the
rigidity of their schemes again possible sources of decoherence and imperfections.
{\bf Response:}We would like to thank the reviewer for their detailed review and valuable suggestions. Regarding the feasibility of the scheme, we have referenced relevant literature (e.g., Phys. Rev. A 78 012312 (2008) and Ann. Phys. 327, 2332 (2012)) in the paper, and we clearly state that our scheme heavily relies on the generation of a 25-qubit entangled state. However, our study is conducted under ideal conditions, so the proposed scheme is theoretically feasible, and we did not specifically address the potential difficulties encountered during the experimental process.
The scheme we propose is based on idealized assumptions and focuses on the construction of quantum communication protocols and the implementation of theoretical models. We acknowledge that in practical applications, factors such as decoherence effects, noise interference, and precision limitations of experimental equipment could affect the realization of the scheme. While these factors may lead to performance deterioration, current technology is capable of addressing these challenges, and our work does not focus on solving these experimental difficulties. Therefore, we did not discuss these implementation challenges in our study.
We appreciate the reviewer pointing this out. We fully understand the impact of noise and decoherence effects on quantum communication systems and recognize their importance in practical operations. However, this paper focuses on the design of idealized protocols and theoretical analysis, so we did not discuss the related issues in experimental implementation. Thank you again for your valuable suggestion.
\textcolor[rgb]{1.00,0.00,0.00}{Comment 5-}There are misprints in this work, e.g., $\sum_{i=1}^4 x_{ki} |j_1j_2\rangle_{X_k \hat{X}_{k’}}$ (page 6) where $x_{ki}$ is independent of $|j_1 j_2\rangle,
“\theta_l,\theta_l,\alpha_l,\alpha_l,…”$ and $\sum_{l=1}^4 |b_{kj}|^2=1 $below Eq. (10),
“quxiliary qubits’ before Eq. (12), “from Eq. (21)” below Eq. (22), “the composite state in
Eq.(21) changes into” below Eq. (24), etc. Please check all the equations and phrasing
carefully.
{\bf Response:}Thank you for your valuable feedback. We have carefully reviewed your comments and made the necessary revisions to address the issues you pointed out. Specifically:
(1) Regarding the typo in the expression $\sum_{i=1}^4 x_{ki} |j_1j_2\rangle_{X_k \hat{X}_{k’}}$, we have modified this expression to ensure clarity and accuracy.
(2)Regarding the repeated symbols in ``$\theta_l, \theta_l, \alpha_l, \alpha_l$..." ,we have corrected the repeated symbols and ensured that the variables are consistently represented throughout the manuscript.
(3) Regarding the issue with $\sum_{l=1}^4 |b_{kj}|^2 = 1$ below Eq. (10), we have fixed the expression in the formula and ensured consistency with the symbols.
(4)Regarding the typo ``quxiliary qubits" before Eq. (12): We have corrected this to ``auxiliary qubits" to ensure proper terminology.
(5)Regarding the phrasing “from Eq. (21)” below Eq. (22) and “the composite state in Eq. (21) changes into” below Eq. (24), we have revised these sections to ensure accurate references to the equations and improved the clarity of the expressions.
We have thoroughly checked the manuscript for any other errors or inconsistencies and have made the necessary revisions. The updated manuscript now accurately reflects these changes.
Thank you again for your careful review and constructive suggestions.
\textcolor[rgb]{1.00,0.00,0.00}{Comment 6-}There are also some repeated sentences in the manuscript, e.g., that in the first paragraph of Sec. 3.1 and in the last paragraph of page 16.
{\bf Response:}Thank you for your careful review and insightful comments. We have addressed the issue of repeated sentences in the manuscript, particularly the repetition in the first paragraph of Section 3.1 and the last paragraph on page 16. These repeated sections have been revised and the unnecessary redundancy has been removed for clarity and conciseness.
We sincerely appreciate your detailed review and valuable comments on our manuscript. We have made the necessary revisions and updates based on your suggestions. Thank you for your feedback and thoughtful guidance, which have been crucial in improving and refining the manuscript. We look forward to your further feedback to help us enhance and perfect the manuscript, aiming for a successful publication.Once again, thank you for your time and effort.
